# Unistrand piRNA clusters are an evolutionarily conserved mechanism to suppress endogenous retroviruses across the *Drosophila* genus

Jasper van Lopik[1,2], Azad Alizada [1], Maria-Anna Trapotsi[1], Gregory J. Hannon [1], Susanne Bornelöv [1,2] ✉ & Benjamin Czech Nicholson [1] ✉

The PIWI-interacting RNA (piRNA) pathway prevents endogenous genomic parasites, i.e. transposable elements, from damaging the genetic material of animal gonadal cells. Specific regions in the genome, called piRNA clusters, are thought to define each species' piRNA repertoire and therefore its capacity to recognize and silence specific transposon families. The unistrand cluster *flamenco* (*flam*) is essential in the somatic compartment of the *Drosophila* ovary to restrict *Gypsy*-family transposons from infecting the neighbouring germ cells. Disruption of *flam* results in transposon de-repression and sterility, yet it remains unknown whether this silencing mechanism is present more widely. Here, we systematically characterise 119 *Drosophila* species and identify five additional *flam*-like clusters separated by up to 45 million years of evolution. Small RNA-sequencing validated these as bona-fide unistrand piRNA clusters expressed in somatic cells of the ovary, where they selectively target transposons of the *Gypsy* family. Together, our study provides compelling evidence of a widely conserved transposon silencing mechanism that co-evolved with virus-like *Gypsy*-family transposons.

Transposable elements (TEs) are widespread across all domains of life. TEs are broadly categorized based on their structure and mobilisation strategies into DNA transposons, which move via a "cut-and-paste" mechanism, and retrotransposons (reviewed in[1]). Retrotransposons replicate via RNA intermediates and are further subdivided into non-LTR elements, including short interspersed nucleotide elements (SINEs) and long interspersed nucleotide elements (LINEs), and long terminal repeats (LTR) elements, which share similarity to endogenous retroviruses (ERVs). LTR transposons and ERVs both encode *gag* and *pol* open reading frames (ORFs), with ERVs and specialised retroelements (also known as errantiviruses) such as *gypsy* and *ZAM* also possessing an *envelope* (*env*) gene. The *env* gene allows virus-like particle formation and cell-to-cell transposition in

addition to the "copy-and-paste" mobilisation mechanism intrinsic to all LTR TEs.

The ability of transposons to mobilise and thereby move or copy from one genomic location to another forms a threat to the genome integrity of their host. This activity, if present in gonadal cells, typically results in sterility[2,3]. Multiple molecular mechanisms have evolved to suppress TE activity, including the HUSH complex, KRAB-zinc finger proteins and the PIWI-interaction RNA (piRNA) pathway[4–9]. The animal-specific piRNA pathway is predominantly expressed in gonadal cells and relies on 23-30 nt small RNAs, mainly derived from transposons and/or discrete genomic loci dubbed piRNA clusters, that associate with PIWI-clade Argonaute proteins[8–10]. While piRNA clusters are widely found throughout the animal kingdom, including examples in

[1]Cancer Research UK Cambridge Institute, University of Cambridge, Li Ka Shing Centre, Cambridge CB2 0RE, UK. [2]These authors contributed equally: Jasper van Lopik, Susanne Bornelöv. ✉e-mail: Susanne.Bornelov@cruk.cam.ac.uk; ben.nicholson@cruk.cam.ac.uk

human, mice, zebrafish and mosquitos[11–14], their function and content varies with not all species showing enrichment of transposon remnants.

In *Drosophila melanogaster*, piRNA clusters are enriched in TE fragments reflecting past and current transposon burden. Depending on their ability to generate piRNAs from one or both genomic strands, piRNA clusters are classified as either unistrand or dual-strand[10]. Dual-strand clusters, as well as the factors ensuring their transcription and export to piRNA processing sites, are expressed specifically in germ cells and appear to be both fast evolving and *Drosophila*-specific[15–17]. In the somatic compartment of the ovary, however, piRNAs are derived mainly from unistrand clusters. Transcripts from unistrand clusters are similar to canonical mRNAs in that they derive from discrete promoters, are spliced, likely polyadenylated, and are exported via the canonical Nxf1-Nxt1 machinery[16,18,19].

The *Drosophila* ovary contains somatic follicle cells that encapsulate and support the germ cells, including the oocyte. Through genetic experiments, the *flamenco* (*flam*) locus was identified as the master regulator of *Gypsy*-family transposons in the follicle cells of *Drosophila melanogaster*, well before the piRNA pathway was discovered[20,21]. Several studies initially attempted to link protein-coding genes in the *flam* region with *Gypsy* repression, but *flam* was eventually found to be a non-coding RNA gene[10] containing numerous LTR TE fragments, from elements such as *idefix*, *ZAM* and *gypsy*, that are predominantly inserted in the same genomic orientation[22]. Following transcription and processing, *flam* gives rise to a diversity of abundant piRNAs with sequence complementarity to these transposons. Loss of *flam* expression or the failure to process its transcripts into piRNAs results in a near complete loss of *Gypsy*-targeting small RNAs in the somatic cells of the ovary, permitting ERV-like elements to form virus-like particles able to infect germ cells and ultimately resulting in sterility[21,23–25]. Thus, the current view in the field is that *flam* acts as a transposon trap[26–29], where a new TE able to mobilise from somatic cells initially will increase its copy number over generations, until it eventually becomes integrated into the *flam* cluster (Fig. 1a), leading to its silencing from the subsequent generation (Fig. 1b).

The indispensable role of *flam* in TE regulation in *D. melanogaster* has sparked questions about its evolutionary conservation within the wider *Drosophila* genus. Although *flam* has been identified in closely related species, it appeared to be absent in *D. ananassae* and two members of the *obscura* group[30,31]. Making use of numerous recently released high-quality *Drosophila* genome assemblies[32,33], here we systematically searched for *flam*-like unistrand piRNA clusters within the *Drosophila* subgenera *Sophophora* and *Drosophila*, revealing their widespread presence. Our results highlight their unique characteristic architecture and specificity in regulating somatically active LTR elements, particularly those carrying an *envelope* protein that facilitates transfer to germ cells. Collectively, our study suggests a conserved and essential role of somatically expressed unistrand piRNA clusters in the suppression of ERVs across the entire *Drosophila* genus.

## Results

### *Flam* is evolutionarily conserved beyond the *melanogaster* subgroup

The unistrand cluster *flam* is the major source of piRNAs in somatic follicle cells of the *D. melanogaster* ovary. *Flam* is characterised by an array of antisense oriented remnants of *Gypsy* family transposons, which gives rise to piRNAs complementary to active TEs[34]. As genome assembly quality improved, the size of the *flam* locus has steadily increased from an original annotation of ~180 kilobases (kb) up to an approximate 650 kb[20,35,36]. Despite its indispensable role in TE control[10,20,21], there has been no evidence of *flam* conservation outside of the *melanogaster* subgroup[30]. We therefore asked whether *flam* conservation could be extended further by analysing a total of 193

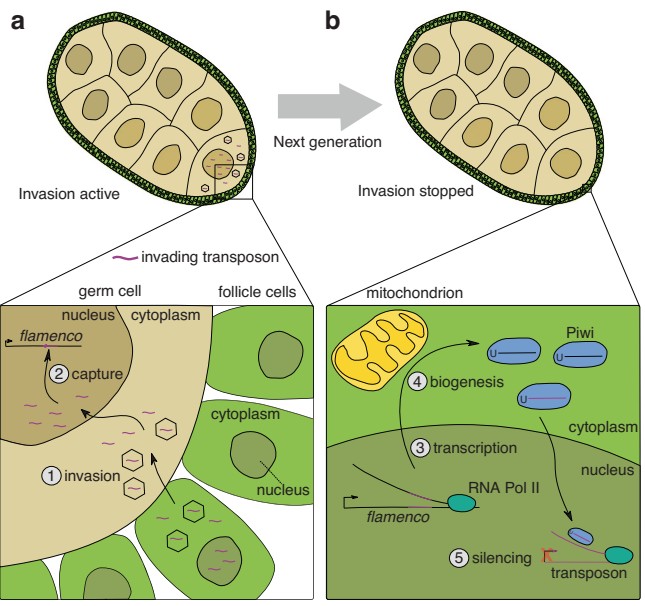

**Fig. 1 | Model of ERV invasion and capture by the *flamenco* piRNA cluster.**
**a** Cartoon of a developing *Drosophila* egg chamber with an active transposon invasion from the soma (top). Somatic follicle cells lining the egg chamber are shown in green and germ cells are shown in beige. Transposon transcripts (purple) originating from somatic cells enter the germ cells (bottom, step 1, invasion). Once reverse transcribed and transported into the nucleus, they integrate into the germline genome. Transposon copy number increases over multiple generations, until a transposon is inserted in antisense orientation into *flam* (step 2, capture). **b** Cartoon of a *Drosophila* egg chamber in which transposon invasion is halted (top). A piRNA precursor transcript is produced from the *flam* locus in somatic cells (bottom, step 3, transcription). The precursor is processed into piRNAs and loaded into Piwi proteins (step 4, biogenesis). The Piwi-piRNA complex enters the nucleus where it recognises transposon transcripts by sequence complementarity and instruments their co-transcriptional repression (step 5, silencing).

Drosophilid genomes from 119 species, including recently published high-quality, long-read genome assemblies[32,33].

We first performed a synteny analysis to locate *flam*-syntenic regions in other species by mapping 20 genes up- and downstream of the *D. melanogaster* cluster to each target genome assembly (Fig. 2a). Extensive accumulation of TE insertions predominantly in one genomic orientation was suggestive of a unistrand cluster at the expected genomic location across nearly all studied species within the *melanogaster* subgroup (Fig. S1). Within the *melanogaster* subgroup, species, where a syntenic *flam* cluster was not apparent, generally had more fragmented genome assemblies. Additionally, *flam*-syntenic candidate clusters were identified in several species outside the *melanogaster* subgroup, including species from the *suzukii* subgroup, but not the *elegans/rhopaloa* subgroups (Fig. 2b–d, Fig. S2). These clusters ranged from 227 kb (*D. biarmipes*) to 1,085 kb (*D. subpulchrella*) in size (Fig. 2e), and, like their *D. melanogaster* counterpart, displayed a clear strand bias with most transposon fragments inserted opposite to the inferred direction of cluster transcription.

In conclusion, the *flam* locus likely appeared between 13.3 and 15.1 million years ago (MYA), following the emergence of the *elegans/rhopaloa* subgroups, and was detected in 12 species (Fig. 2e). The absence of *flam* beyond these species, despite largely conserved gene synteny in the region and the widespread presence of *Gypsy*-family elements in Drosophilids, prompted the question whether analogous *flam*-like unistrand piRNA clusters exist elsewhere in the genomes of these species.

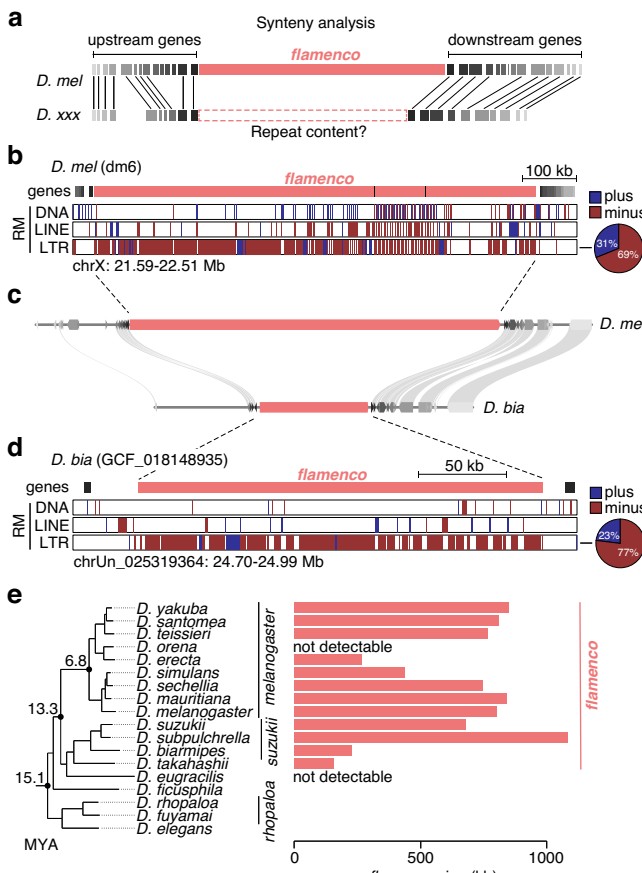

**Fig. 2 | Identification of *flam* across *Drosophila* species. a** Cartoon showing synteny analysis pipeline. **b** Genome browser tracks showing the *D. melanogaster* (dm6 genome) *flam* region with transposon annotation by RepeatMasker (RM), displaying some neighbouring genes used for synteny analysis. The pie chart to the right indicates LTR content per strand in the cluster region. **c** MCScan plot showing gene and *flam* synteny between *D. melanogaster* and *D. biarmipes*. **d** As in (**b**) but showing the *D. biarmipes* (GCF_018148935 genome) *flam* region. **e** Phylogenetic tree (left) representation of the *melanogaster, suzukii, takahashii, eugracilis, ficusphila, rhopaloa* and *elegans* subgroups, indicating the respective size of their *flam*-syntenic loci in kb (right). Abbreviations: MYA million years ago. Source data are available in the source data file.

## D. ficusphila possesses a *flam*-like piRNA cluster

Interspersed repeat content was recently estimated across Drosophilids[32], including the subgenera *Drosophila* and *Sophophora* (to which *D. melanogaster* belongs). We noticed that species of the *Drosophila* subgenus generally appeared to have less repeat content compared to those belonging to the *Sophophora* subgenus. We hypothesised that database-driven repeat annotation commonly performed by RepeatMasker underestimates the transposon abundance in less well studied species. Therefore, we constructed de novo TE annotations using Extensive de novo TE Annotator (EDTA)[37], and found that this indeed improved repeat annotations (Fig. S3a).

The species most closely related to *D. melanogaster* clearly lacking a unistrand syntenic *flam* cluster is *D. ficusphila*. Interestingly, with our novel TE annotations, we found a region enriched in LTR elements at the *flam*-syntenic location, however, without any orientation bias of the transposon insertions (Fig. 3a, b). RNA-seq and small RNA-seq (sRNA-seq) from ovaries revealed that this locus is transcribed and that its transcripts are processed into piRNAs. These piRNAs emanate from both genomic strands and show a 1U bias characteristic to this class of small RNAs (Fig. 3b, c). Overall, this pattern of piRNA production and the organisation of the TE insertions strongly resembles the

architecture of a dual-strand piRNA cluster. The production of piRNAs from dual-strand clusters in *D. melanogaster* germ cells is amplified by the germline-specific ping-pong cycle[10,38], which is characterised by the presence of complementary piRNA pairs overlapping by precisely ten nucleotides counted from their 5′ ends. Ping-pong and phasing analysis on piRNAs uniquely derived from the *flam*-syntenic cluster in *D. ficusphila* revealed phasing (Fig. S3b) and a strong ping-pong signature (Fig. 3d), indicating that this piRNA cluster is likely expressed and processed in the germline.

As *D. ficusphila* appears to possess a dual-strand cluster in place of the *flam* locus, it either lacks somatically expressed LTR transposons or controls these TEs by other means. The presence of *Gypsy* family elements in all investigated genomes strongly indicates that *D. ficusphila* has somatically expressed transposons (Fig. S3c). We therefore set out to identify non-syntenic unistrand piRNA clusters in *D. ficusphila* that resemble *flam* in terms of its size, *Gypsy*-family TE content and strong enrichment for transposon insertions to be oriented on one genomic strand.

We calculated LTR transposon content across 100 kb sliding windows to scan the entire genome for putative unistrand piRNA clusters (Fig. 3a). This identified a ~560 kb region enriched in LTR TEs predominantly on the plus strand of chrUn_025064091 (Fig. 3a, e), hereafter referred to as *flamlike1*. Whole ovary sRNA-seq revealed that *flamlike1* produces piRNAs complementary to TE transcripts, thus resembling the expression pattern of a unistrand cluster. Accordingly, piRNAs mapping to this locus also show a strong 1U bias, robust phasing pattern and a weaker ping-pong signature (Fig. 3f–g, Fig. S3d).

The presence of a dual-strand cluster at the *flam*-syntenic region and the identification of a *flam*-like unistrand cluster elsewhere in the genome prompted us to investigate whether *D. melanogaster* also has a piRNA cluster at the region syntenic to *flamlike1*. A macrosynteny analysis between *D. ficusphila* and *D. melanogaster* indicated that the *flamlike1*-syntenic region is located on chromosome 2 L in *D. melanogaster* (Fig. 3i). Closer investigation revealed this to be a purely genic region bearing no TE enrichment, whereas the *flam*-syntenic region shows a piRNA cluster in *D. ficusphila* (Fig. 3h–j).

## *Flam*-like clusters repeatedly emerged throughout evolution

The observation of a *flam*-like cluster in *D. ficusphila* raises the question if other, more distant species also carry *flam*-like clusters and whether this is an evolutionary conserved genome feature. To determine whether *flam*-like loci can be readily identified across Drosophilids, we applied the genome-wide scanning approach to all 119 species, including *D. melanogaster*. This re-identified *flam* and several *flam*-syntenic loci across nine species of the *melanogaster* group, including *D. suzukii, D. takahashii* and *D. biarrmipes* (Fig. S4). Publicly available and our own genomics data confirmed that these loci are expressed and produce abundant piRNAs (Fig. S5).

Interestingly, we also identified five additional *flam*-like loci outside of the *melanogaster* group in *D. oshimai, D. persimilis, D. pseudoobscura, D. innubila,* and *D. bifasciata* (Fig. S6). Of note, *D. persimilis* and *D. pseudoobscura* are closely related species that diverged less than 2 MYA, and synteny analysis revealed that their *flam*-like loci were syntenic (Fig. S7a). We named these loci *flamlike2, flamlike3* (both *D. persimilis* and *D. pseudoobscura*), *flamlike4,* and *flamlike5,* respectively (Fig. 4a–d). Publicly available sRNA-seq data available for *D. persimilis* and *D. pseudoobscura* revealed that *flamlike3* produces vast amounts of piRNAs that predominantly originate from one genomic strand and were in antisense orientation to LTR transposon transcripts (Fig. S7b), which was also confirmed by our own sRNA-seq data (Fig. 4b, Fig. S7b). Similarly, our sRNA-seq data confirmed the production of piRNAs from *flamlike5* in *D. bifasciata* (Fig. 4d). Notably, *flamlike3* in *D. pseudoobscura* and *flamlike5* in *D. bifasciata* correspond to soma-expressed piRNA clusters observed in a recent publication[39].

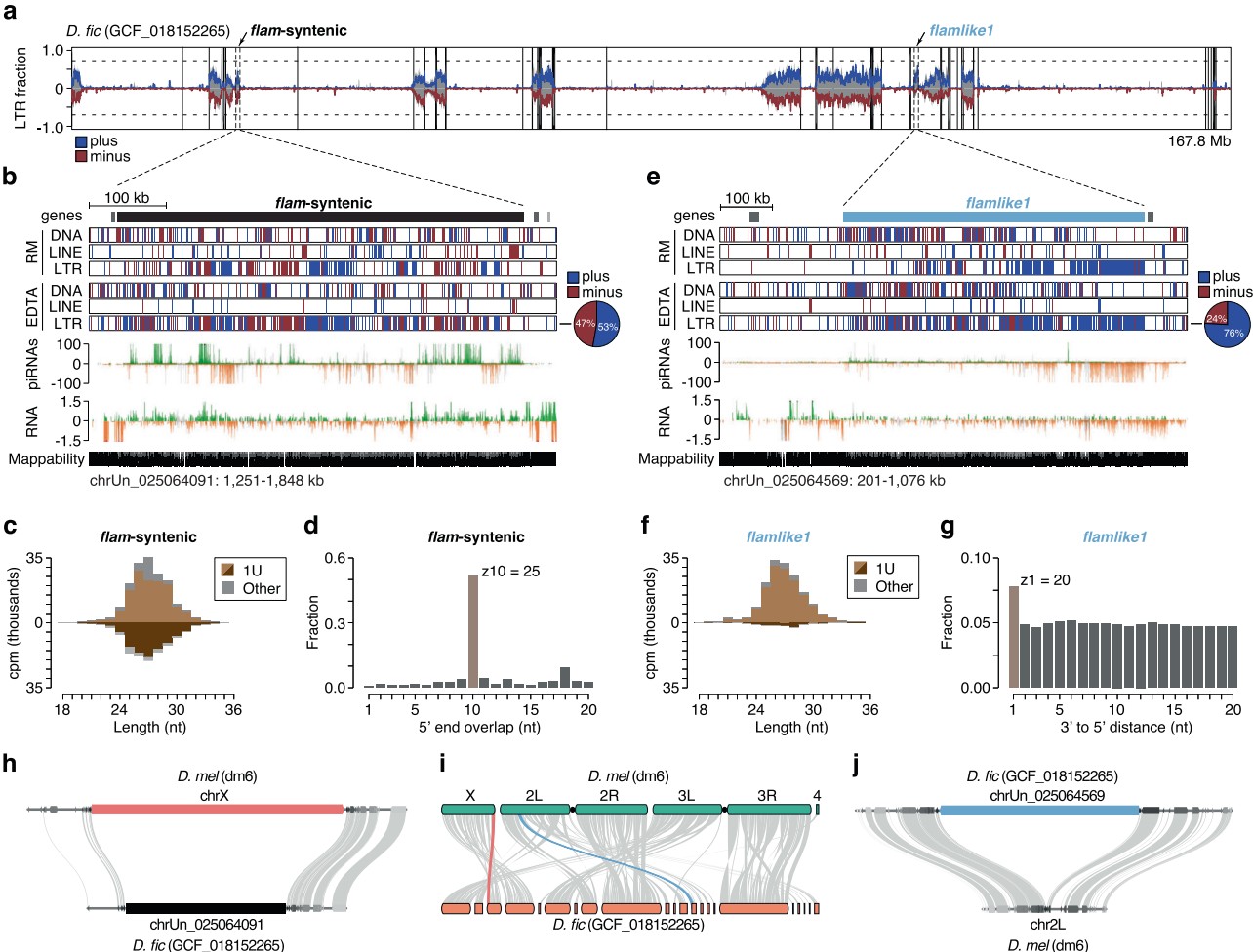

**Fig. 3 | Identification of a non-syntenic *flam*-like locus in *D. ficusphila*.**
**a** Genome-wide detection of *flam*-like loci in *D. ficusphila* using a sliding window approach. Predicted LTR content (blue, plus strand; red, minus strand) and total repeat content (grey) is shown across the whole genome (100 kb bins). Arrows indicate the *flam*-syntenic region, and the de novo identified *flamlike1* region. **b** Genome browser tracks showing *D. ficusphila* (GCF_018152265 genome) *flam*-syntenic region (black bar) with transposon annotations by RepeatMasker (RM) and EDTA, and gene annotation by NCBI. Uniquely mapping piRNA (cpm) and total RNA levels (ln(cpm+1)) are presented (green/orange, unique mappers; grey, multi-mappers). Mappability is displayed at the bottom. The pie chart to the right indicates LTR content per strand in the cluster region. **c** Relative piRNA size distribution of piRNAs mapping sense (light brown) and antisense (dark brown) to *D. ficusphila flam*-syntenic region. **d** Ping-pong signature for piRNA pairs mapping onto the *flam*-syntenic region. **e** Same as in (b), but for *D. ficusphila flamlike1* region (blue bar). **f** Relative piRNA size distribution of sense (light brown) and antisense (dark brown) piRNAs mapping to *D. ficusphila flamlike1* region. **g** Phasing signature (3' end to 5' end distance) for piRNAs mapping onto *flamlike1*. **h** Zoom-in on genic region indicating presence of a piRNA cluster in the *flam*-syntenic region in *D. ficusphila*. **i** Macrosynteny plot indicating gene synteny between *D. melanogaster* and *D. ficusphila* highlighting *flam* (red) and *flamlike1* (blue). **j** Zoom-in on genic region indicating the absence of a piRNA cluster in *flamlike1*-syntenic region in *D. melanogaster*. Source data are available in the source data file.

We noted that several *flam*-syntenic regions escaped detection through the genome-wide scanning approach in highly fragmented genome assemblies (Fig. S8). This observation, together with the presence of syntenic *flam*-like clusters in the closely related species *D. persimilis* and *D. pseudoobscura* raised the possibility that additional syntenic loci might also be present for other candidate unistrand clusters. Synteny analysis of *flamlike3* showed a widespread conservation of this locus within the *pseudoobscura* subgroup, being present in at least five species (Fig. 4e, Fig. S7b, Fig. S9a–d). Similarly, for another group of four species within the *obscura* subgroup we identified *flamlike5* (Fig. 4d, Fig. S9d–g, Fig. S10). Interestingly, *flamlike5* is syntenic to a dual-strand cluster in the species carrying *flamlike3* (Fig. 4f, Fig. S9a, d, Fig. S11). Similar to the *flam*-syntenic region in *D. ficusphila*, these *flamlike5* syntenic dual-strand clusters are enriched for LTR transposons, but do not show an orientation bias. Thus, large *flam*-like unistrand clusters appear to have emerged at various branches of the *Drosophila* genus (Fig. 4e, Supplementary Data 1).

## *Flam*-like clusters are expressed in the somatic follicle cells of the ovary

Identification of unistrand piRNA clusters across 22 *Drosophila* species (excluding *D. melanogaster*), all displaying similar genomic characteristics, raised the hypothesis that these loci have a somatic function similar to that of *flam*. Supporting this, several of these loci were confirmed to be transcribed predominantly from one strand (Fig. 4b, d, Fig. S7b), consistent with expression in somatic cells that, in *D. melanogaster*, lack the machinery needed to express and export transcripts from dual-strand clusters. However, since sRNA-seq from whole ovaries captures a mixture of both somatic and germline piRNAs, it remained uncertain if these unistrand piRNA clusters actually operate in the soma.

To determine whether *flam*-like loci are somatically expressed, we generated and sequenced sRNA-seq and RNA-seq libraries that were enriched for somatic follicle cells of the ovary (Fig. 5a) from ten and five species, respectively, including species carrying the *flam*-syntenic,

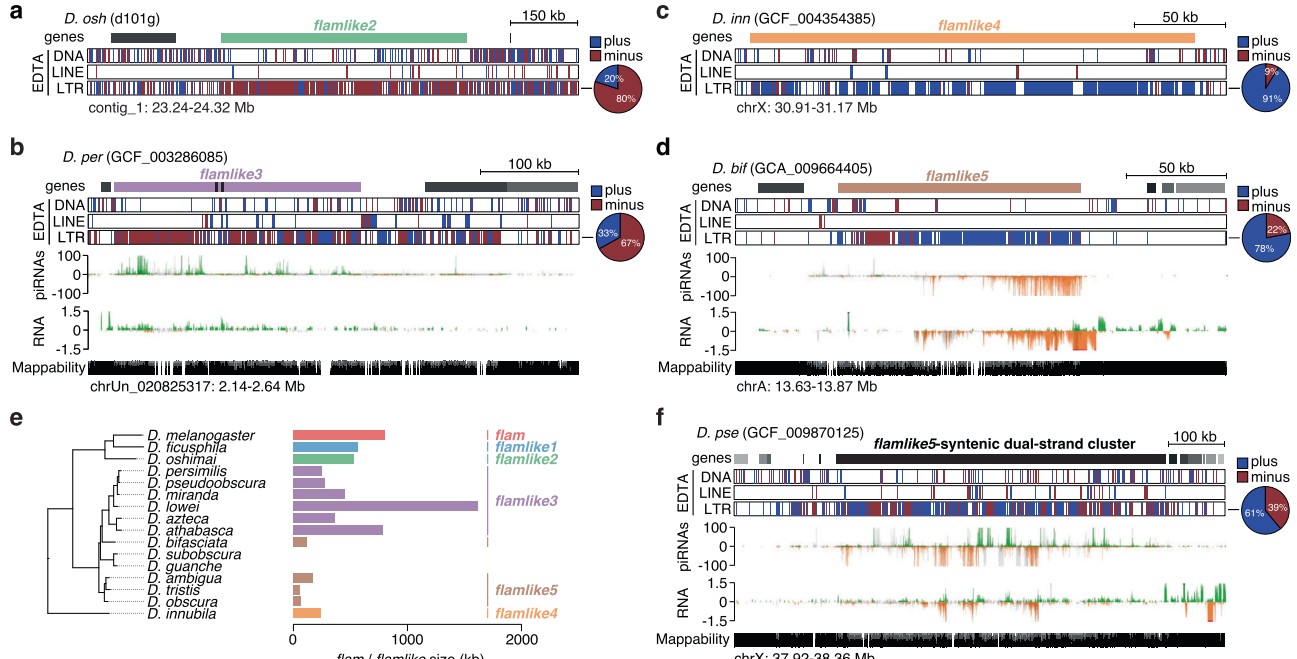

**Fig. 4 | Multiple *flam*-like clusters are identified across diverse *Drosophila* species. a** Genome browser tracks showing *D. oshimai flamlike2* region with transposon annotation by EDTA (blue, plus strand; red, minus strand) and gene annotation by NCBI. The pie chart to the right indicates LTR content per strand in the cluster region. **b** As in (**a**) but showing the *D. persimilis flamlike3* region. Uniquely mapping piRNA (cpm) and total RNA levels (ln(cpm+1)) are presented (green/orange, unique mappers; grey, multi-mappers). Mappability is displayed at the bottom. **c** As in (**a**) but showing the *D. innubila flamlike4* region. **d** As in (**b**) but showing *D. bifasciata flamlike5* region. **e** Phylogenetic tree representation highlighting *flamlike1*, *flamlike2* and *flamlike4* presence and *flamlike3* and *flamlike5* conservation across *Drosophila* species, with *D. melanogaster flam* as a reference (left). Cluster size is indicated in kb (right). **f** As in (**d**) but showing the *D. pseudoobscura flamlike5* syntenic dual-strand cluster. Source data are available in the source data file.

*flamlike1*, and *flamlike3* piRNA clusters. To allow precise mapping of the putative promoter regions, we further generated ATAC-seq libraries from nine of these species. Based on our RNA-seq and ATAC-seq libraries, we first refined the predicted location of the transcription start site (TSS) for our *flam*-syntenic and *flam*-like clusters across 12 species (Supplementary Data 1). Cross-species analysis of ATAC-seq peaks further confirmed that syntenic clusters share orthologous open chromatin regions of the promoter area (Fig. S12, S13). For *flam*-syntenic clusters, we identified several conserved *cis*-regulatory elements in their promoter peaks (Fig. S12), including the reported Cubitus interruptus (Ci) binding site and Initiator (Inr) element[18]. These elements and peaks were conserved from *D. melanogaster* to *D. yakuba* (Fig. S12) but were not readily detected in more distantly related species (Fig. S12, Fig. S13). Nevertheless, the presence of conserved regulatory regions suggests that the promoter regions are under purifying selection and that unistrand clusters are expressed as canonical transcription units.

Remarkably, both *flam*-syntenic, *flamlike1*, and *flamlike3* displayed a strong piRNA strand bias (between 7- and 70-fold) (Fig. 5b, Fig. S14a) and produced a greater fraction of piRNAs in somatic cells as compared with whole ovaries (Fig. 5c–d).

Interestingly, we found that *D. yakuba* and *D. erecta* deviated from this pattern, displaying somatic expression at the 5′ end and germline expression towards the 3′ end of the *flam*-syntenic region (Fig. S14b, c). Together, this indicates that all identified *flam*-like loci produce antisense piRNAs capable of targeting transposons and that they are expressed primarily in the somatic follicle cells of the ovary.

### Somatic unistrand piRNA clusters may be universal across *Drosophila*

After identifying *flam*-like clusters in many but not all species, we proceeded to investigate whether the absence of *flam*-like clusters in

the remaining species indicate that a somatic piRNA pathway is absent in some species or if our current approach, solely relying on TE annotations, is insufficient to detect them. To test this, we generated total and soma enrichment sRNA-seq libraries for *D. ananassae*, *D. mojavensis*, and *D. virilis*, representing three distinct groups across the *Drosophila* genus. These libraries, together with our previously generated sequencing data, were used to identify major piRNA-producing loci (>35 kb) using proTRAC[40] across 12 species. Up to 14 clusters were identified per genome assembly (mean 7.4 ± 3.7, standard deviation) and classified as somatic, intermediate, or germline based on their somatic expression ratio (Supplementary Data 2). Somatic expression was strongly associated with strand bias, both across *D. ananassae*, *D. mojavensis*, and *D. virilis* (Fig. 5e, f, Fig. S14d, e) as well as the species investigated originally (Fig. S15), except for *D. erecta* and *D. yakuba* as these have both a somatic and a germline component arising from the same genomic location. Notably, these somatic piRNA clusters were all enriched in LTR transposons, suggesting that they might play a conserved role in the repression of these TEs in the soma (Fig. 5d, f, Fig. S14d, e).

The sRNA-seq-assisted identification of somatic *flam*-like piRNA clusters in three additional species (Fig. 5e) suggests that somatic unistrand piRNA clusters may be a universal feature across *Drosophila* (Fig. 5g). In support of a *flam*-like role, we observed that these newly-identified clusters also displayed pericentromeric localisation (Fig. S16a–c) and at least one of these cluster appears to be conserved across a much wider group of species (Fig. S16d–g). Together, this indicates a strong selective pressure to maintain production of transposon-targeting piRNAs in somatic follicle cells.

### Unistrand *flam*-like clusters control ERVs in somatic follicle cells
The canonical function of the piRNA pathway is arguably the suppression of parasitic elements in gonadal cells. At some point, an

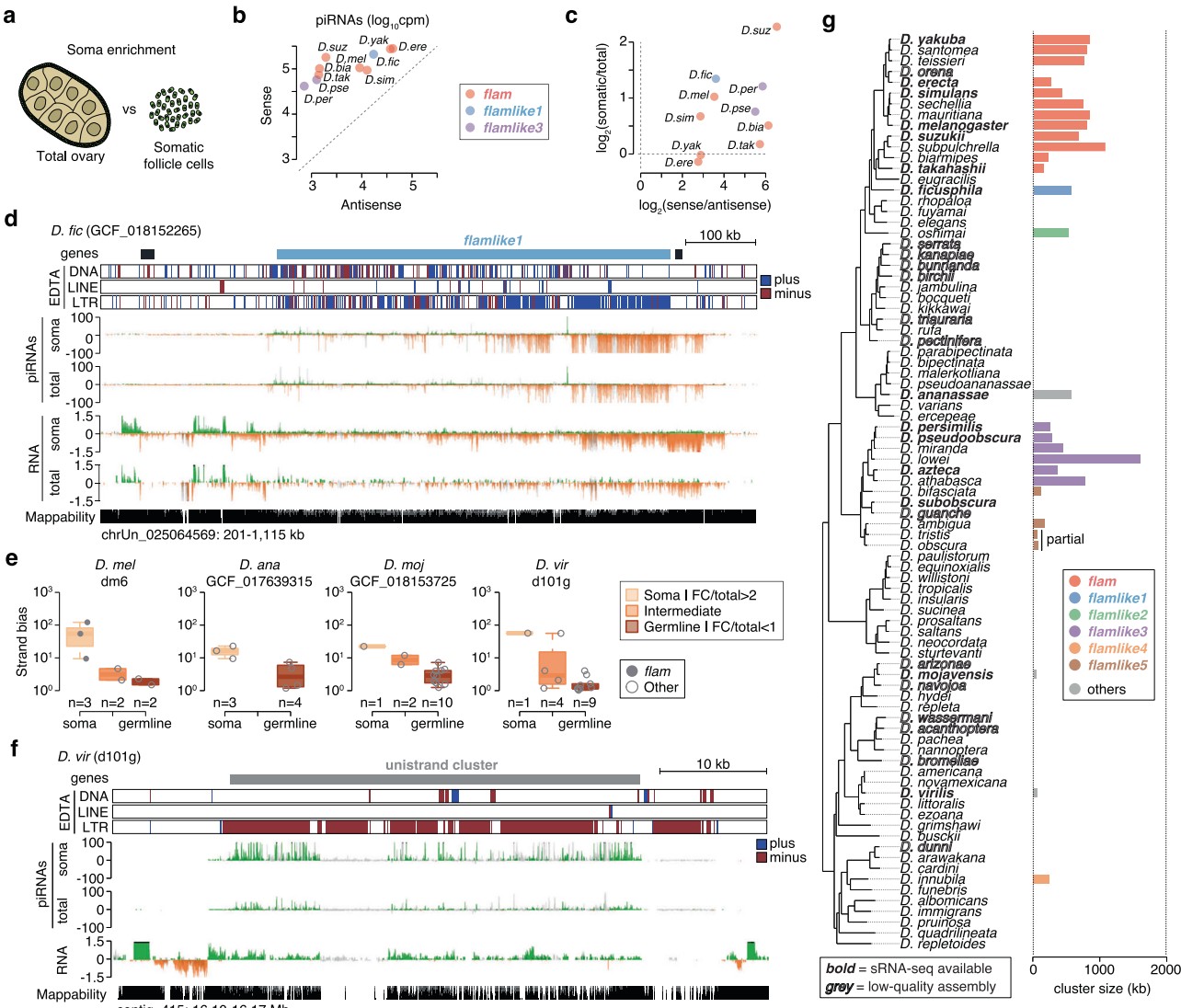

**Fig. 5 | *Flam*-like and unistrand piRNA clusters are somatically expressed.**
**a** Illustration of cell composition in total ovary (germline cells surrounded by somatic follicle cells) or after somatic follicle cell enrichment, respectively. **b** Soma enrichment library piRNAs mapping to piRNA clusters in sense or antisense orientation across 9 species. The largest cluster representative is shown for species with multiple genome assemblies. For all clusters see Fig. S14a. **c** Scatterplot of piRNA strand bias against piRNA soma enrichment across the indicated piRNA clusters (9 species). The largest cluster is shown for species with multiple genome assemblies. For all clusters see Fig. S14b. **d** Genome browser tracks of *flamlike1* in *D. ficusphila* showing transposon annotations (EDTA), soma-enriched piRNAs (cpm), total piRNAs (cpm), soma-enriched RNA expression (ln(cpm+1) scale), and total RNA expression (ln(cpm+1) scale), and mappability. Transposon annotations are shown in red (minus strand) or blue (plus strand). Sequencing data is shown in

green or orange for uniquely mapped reads, and grey for multi-mapping reads. **e** Analysis across de novo identified large piRNA clusters. Clusters were classified as soma, intermediate or germline based on the follicle cell versus total ovary piRNA ratio. Strand bias ($\log_{10}$ scale) is shown across each category. Boxplots show median (central line), interquartile range (IQR, box), and minimum and maximum values (whiskers, at most 1.5*IQR). Data points represent individual clusters. All species shown in Fig. S15. **f** As in (**d**) but showing a somatic piRNA cluster in *D. virilis*. See also Fig. S14e. **g** Phylogenetic tree summarising all studied somatic piRNA clusters across all analysed species ($n = 119$). Cluster size represents the mean across all assemblies, except in the "others" category, where a single cluster is shown. Species names in bold have sRNA-seq data to validate their expression. Abbreviations: cpm counts per million, FC follicle cell. Source data are available in the source data file.

ancestral LTR transposon of the *Gypsy* family obtained an *env*-like ORF, likely from an insect baculovirus, thereby gaining properties of an endogenous retrovirus (ERV)[41–43]. This allowed the TE, in addition to its ability to mobilize across the genome, to move from cell to cell and infect the oocyte from the surrounding somatic follicle cells[29,41,42]. *D. melanogaster* counters these ERVs through a somatic piRNA pathway in conjunction with expression of the *flam* locus specifically in this tissue[23,44,45]. Whether the silencing of ERVs by somatic piRNAs is conserved in other species is currently unknown, though it has been noted that the infectious properties of the *env* ORF are present and appear repressed in the *obscura* group[43].

Presently characterised *Gypsy* family TEs are predominantly from *D. melanogaster* and therefore insufficient to use for analysis of distantly related *Drosophila* species. To enable an unbiased analysis, we constructed curated transposon consensus sequences for all 193 analysed genome assemblies (Fig. S17). Briefly, raw EDTA and RepeatModeler libraries were combined, filtered and deduplicated to retain a minimal set of TE consensus sequences (see Methods, Construction of curated de novo transposon libraries). These were annotated with respect to their open reading frames (ORFs), repeat classification, and genomic distribution (Supplementary Data 3). Using our TE libraries, we identified genome repeat content of up to 55% (Fig. S18), extending

previous estimates based on RepeatMasker annotations[32]. Our analysis further revealed a strong correlation between genome assembly size and interspersed repeat content (Fig. S19a). LTR elements dominated the repeat landscape across most species (Fig. 6a, Fig. S19b–d), with the *Gypsy* family generally most abundant, representing about 25% of all interspersed repeats (Fig. S19d). The *Gypsy* family also had the highest number of subfamilies (Fig. S20, Supplementary Data 4) and genomic copies (Fig. S21, Supplementary Data 5). As expected, an *env* ORF was frequently found in *Gypsy* elements (Fig. 6b), but not in the other LTR families *Copia* and *Pao*. Notably, *env* ORFs were found in *Gypsy* elements across all species (Fig. S19e), suggesting that its acquisition is ancestral to the *Drosophila* genus. It is therefore likely that *Gypsy* ERVs mobilise from somatic follicle cells into the germline across all *Drosophila* species.

Next, we compared our curated TE libraries to 180 well-characterised *Drosophila* transposon subfamilies (Fig. S22, Supplementary Data 6), revealing that many were present in multiple species. Focusing on TEs well-known in *D. melanogaster*, we found convincing evidence of vertical transmission reflecting their phylogenetic relationship within the *melanogaster* subgroup, but also evidence of horizontal transfer to *D. ercepeae*, *D. bocqueti*, *D. pruinosa*, and the

*Zaprionus* genus (Fig. S22). Importantly, these observations were supported by cross-species alignment of our sRNA-seq data (Fig. S23), suggesting that similarities in piRNA populations between species is indicative of shared transposon burden.

Analysis of repeat content across all *flam*-like loci revealed a strong enrichment for LTR transposons inserted in antisense orientation (Fig. 6c). This is expected, since an LTR TE enrichment and strand bias was part of the criteria used to define the loci. However, when analysing the LTR transposon families, we observed that this strand bias was driven exclusively by *Gypsy* elements and absent for other LTR transposons (Fig. 6c). These unistrand clusters thus produce piRNAs specifically targeting *Gypsy* elements in follicle cells. We next asked whether this *Gypsy*-targeting is unique to *flam*-like clusters. As an initial control, we extended the analysis to several well-characterised germline clusters in *D. melanogaster* and our previously identified dual-strand clusters that are syntenic to *flam*, *flamlike1*, *flamlike3*, or *flamlike5*. Both groups displayed substantially lower *Gypsy* content and reduced strand bias (Fig. 6d, Fig. S19f). To expand the analysis to all species with a *flam*-syntenic or *flam*-like cluster, we next compared our *flam*-like clusters to proTRAC-derived de novo clusters (>35 kb). We quantified strand bias as the difference in TE content between the

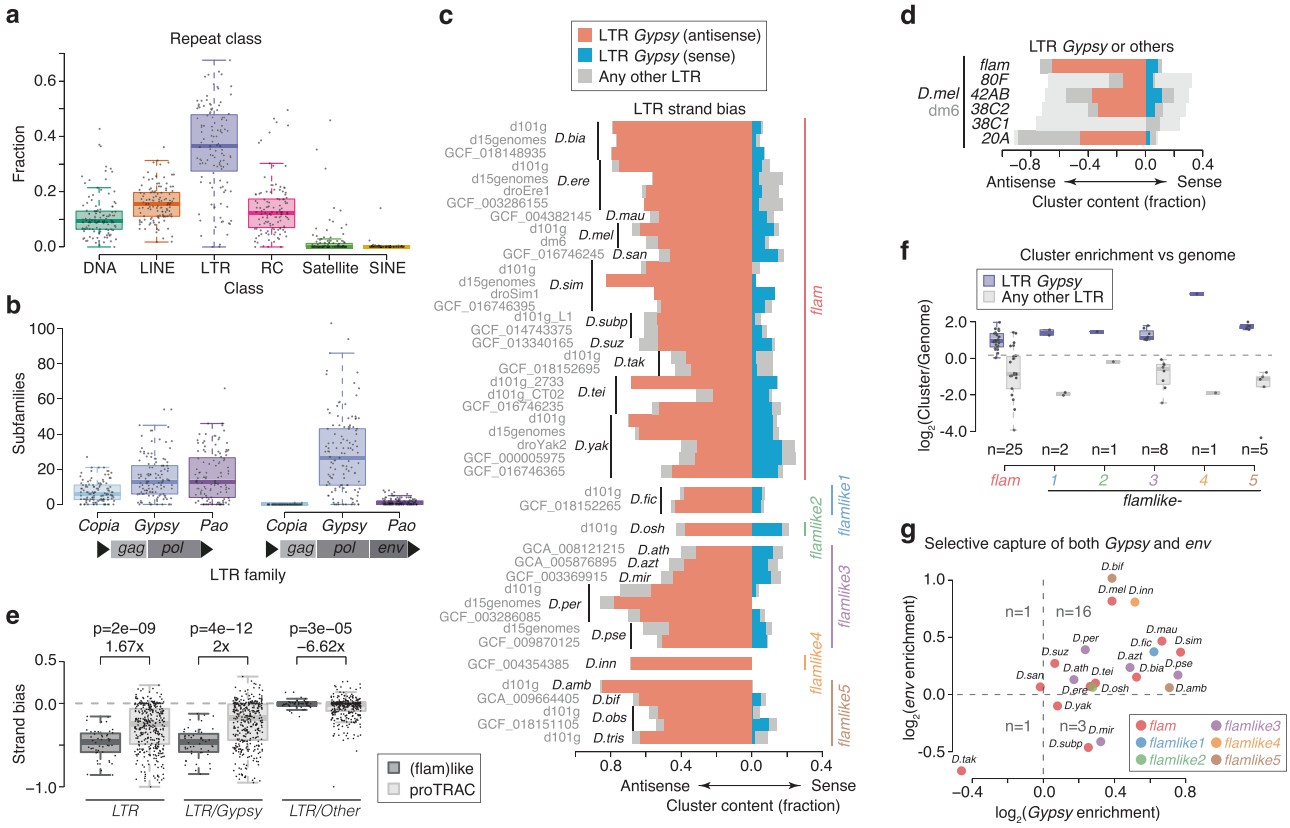

**Fig. 6 | Unistrand *flam*-like clusters are selectively enriched for antisense ERVs.** **a** Boxplot showing fraction of interspersed repeat content for the indicated repeat classes. Each data point represents one species (*n* = 119). Species with multiple genome assemblies are represented by their mean. **b** Boxplot showing the number of subfamilies detected per LTR family with either *gag* + *pol* (left) or *gag* + *pol* + *env* (right) ORFs. Each data point corresponds to one species (*n* = 119). Species with multiple genome assemblies are represented by their mean. **c** Bar plot showing LTR contribution (left, antisense; right, sense) to total transposon content across all annotated *flam*-like clusters. *Gypsy* elements are shown in red (antisense) or blue (sense) and other LTR elements are shown in grey. Clusters are grouped by synteny as indicated to the right. Species and genome assembly (alphabetically sorted) are indicated to the left. **d** Similar to (**c**), but showing LTR content across *flam* and major dual-strand clusters in *D. melanogaster*. Cluster strand was defined according to total transposon content (light grey). **e** Boxplot showing strand bias defined as

sense strand minus antisense strand contribution to total transposon content for transposons classified as LTR, LTR/Gypsy or any other LTR, respectively. Strand bias is shown across all annotated *flam*-like clusters (left, dark grey, *n* = 48) or major dual-strand clusters in *D. melanogaster* and proTRAC de novo predicted clusters (right, light grey, *n* = 354). The means were compared using a two-sided Student's *t* Test. **f** Boxplot displaying *Gypsy* versus other LTR coverage against the genomic average across different unistrand clusters. Each point corresponds to one cluster in one genome assembly. **g** Scatterplot showing *Gypsy* enrichment against *env* enrichment in unistrand clusters from the indicated species (see "Cluster content analyses" in the Methods for details). Only high-quality LTR transposons are included in the analysis (both *gag* and *pol* and at least one good genomic hit). Boxplots show median (central line), interquartile range (IQR, box), and minimum and maximum values (whiskers, at most 1.5*IQR). Source data are available in the source data file.

sense and antisense cluster strands and observed 1.65-fold lower LTR transposon strand bias in proTRAC clusters compared with *flam*-like ones (Fig. 6e, left panel). Strikingly, this reduction was entirely driven by a 1.97-fold reduction in *Gypsy* transposon strand bias (Fig. 6e). Thus, an antisense arrangement of *Gypsy* transposons appears to be a property specific to *flam*-like clusters. To determine whether *Gypsy* family elements are also enriched in these loci, we compared the cluster composition against the genome-wide coverage per LTR family, observing a strong enrichment of *Gypsy* transposons while other LTRs were simultaneously depleted (Fig. 6f).

Finally, to determine whether the unistrand clusters specifically capture *Gypsy* elements, we analysed our TE libraries at subfamily level (Fig. 6b). For this, we calculated a *Gypsy* enrichment ratio as the conditional probability that an LTR transposon captured by the locus was a *Gypsy*-family element, divided by the probability that an LTR element not captured by the cluster was a *Gypsy* family transposon. In total, 19 out of 21 unistrand clusters, including *D. melanogaster flam*, displayed a positive enrichment, indicative of a selective capture of *Gypsy* TEs over other LTR transposons (Fig. 6g, Fig. S19g). Notably, some *Gypsy* subfamilies either never gained or subsequently lost their *env* ORF (Fig. 6b). Based on the expression pattern of non-*env* TEs in *D. melanogaster*[29], these TEs are generally expected to operate in the germline. One such example is *Burdock* in *D. melanogaster*, which shares origin with other ERVs, but has lost its *env* and is now expressed exclusively in the germline[31,42]. This suggests that only transposons possessing an *env* ORF should be controlled by a somatic piRNA cluster. To test this model, we calculated an *env* enrichment ratio, defined as the conditional probability that a *Gypsy* TE captured by the cluster had the *env* ORF present, divided by the probability that a *Gypsy* not captured possessed an *env* domain. This analysis revealed a selective capture of *env*-containing *Gypsy* elements across 17 out of 21 unistrand clusters (Fig. 6g, Fig. S19h), including *D. melanogaster flam*. We note that this analysis required high-quality transposon consensus sequences and therefore some of the exceptions may be due to annotation artefacts. However, we speculate that the *flam*-syntenic region in *D. takahashii*, showing neither *Gypsy* or *env* enrichment and displaying limited soma enrichment (Fig. 5c) may be in the process of losing its *flam*-like function and converting into a germline-expressed dual-strand piRNA cluster.

## Both *flamlike1* and *flamlike3* control soma-expressed *Gypsy* elements

To further our understanding of how transposons are regulated by *flam*-like clusters, we characterised the individual transposons that are controlled by each cluster. For this analysis, we focused on *flam* in *D. melanogaster*, *flamlike1* in *D. ficusphila*, and *flamlike3* in *D. persimilis* and *D. pseudoobscura*. These species were selected based on the availability of both whole ovary and soma-enriched sRNA-seq and RNA-seq. As controls we used the dual-strand *42AB* in *D. melanogaster*, *flam*-syntenic in *D. ficusphila* and *flamlike5*-syntenic in *D. persimilis* and *D. pseudoobscura*.

We first confirmed that the control clusters ($z_{10}$ scores between 21 and 120), but not *flam*-like clusters ($z_{10}$ scores between −0.09 and 0.64), displayed a strong ping-pong signature (Fig. S25a), indicative of an active piRNA amplification pathway that operates in germ cells but not in the soma, as reported for *D. melanogaster*[10,38]. Accordingly, comparison of cluster expression in soma-enriched and whole ovary RNA-seq libraries indicated that *flam*-like clusters are preferentially expressed in the soma (Fig. S25b).

To gain more insight into each cluster, we next analysed transposon subfamilies with high potential to be regulated by the piRNA pathway. To allow for an unbiased analysis, we selected 100 subfamilies based on piRNA abundance in either soma-enriched samples or in whole ovary libraries. These two approaches yielded nearly identical results and allowed us to investigate 116-128 transposons per

species. Since only piRNAs with sequence complementary to a transposon transcript have the potential to repress it, we next characterised the genomic origin of piRNAs mapping antisense to each transposon. For this analysis, we mapped antisense piRNAs to the corresponding genome assembly and assessed their overlap with piRNA cluster regions. We observed that antisense piRNAs were exclusively derived from the sense strand of *flam* and *flam*-like clusters, whereas they originate from both strands of the control clusters (Fig. 7, boxplots). Moreover, and most strikingly, many individual transposons were almost exclusively controlled by a single *flam* or *flam*-like cluster, as indicated by a high proportion of all antisense piRNAs mapping to that cluster. In contrast, the control group containing dual-strand clusters produced piRNAs against more transposon subfamilies, albeit at a lower level, potentially indicating redundancy with other germline clusters (Fig. 7, boxplots).

Arranging the individual transposons by their soma-enrichment revealed a marked difference in the expression profile of transposons regulated by *flam*-like clusters and dual-strand control clusters (Fig. 7, bar graphs). Moreover, several transposons were identified as exclusively controlled (>90% of antisense piRNAs) by a single cluster. Nearly all transposons exclusively controlled by *flam* and *flam*-like clusters were predicted to be *Gypsy*-family transposons (18 LTR/Gypsy, 1 Unknown, 1 DNA/Maverick), whereas a diverse set of families were identified as exclusively controlled by a dual-strand cluster (7 LTR/Gypsy, 3 LTR/Copia, 2 LINE/I-Jockey, 2 LINE/R2, 2 Unknown, 1 DNA/Maverick, 1 LINE/CR1, 1 LINE/I, 1 LTR/Pao, and 1 RC/Helitron). Notably, five of the TEs controlled by dual-strand clusters corresponded to *Circe*, *invader6*, and *BS* in *D. melanogaster*, which are known to be germline expressed, whereas somatically or intermediately expressed *Tabor*, *gypsy3* and *gypsy10* were among the *flam*-controlled hits (Fig. 7, bar graphs). Additionally, the LINE elements *spock* and *worf* originally identified in *D. miranda*[46] were both found to be exclusively controlled by *flamlike5*-syntenic in *D. persimilis* (Fig. 7h).

Together, our analyses find that unistrand *flam*-like piRNA clusters selectively acquired *env*-containing *Gypsy* family transposable elements in antisense orientation and that these in many cases are the sole source of piRNAs against these transposons. In all, our study revealed a conserved role for the piRNA pathway in controlling ERVs in follicle cells across the *Drosophila* genus. These data support the model where somatic *flam*-like piRNA clusters act as a trap for ERV-like elements with TEs able to mobilise outside the germline eventually becoming silenced upon integration into one of these loci (Fig. 1).

## Discussion

New TE insertions can only propagate to the next generation when they are established in the germ cells. Acquisition of a retroviral *env* ORF by an ancestral *Gypsy* family retrotransposon is believed to have transformed it into an endogenous retrovirus. Thereby it gained the means of invading the oocyte from the somatic follicle cells of the ovary, evading silencing by the branch of the piRNA pathway that is germ cell-specific (Fig. 1). In *D. melanogaster*, a somatic piRNA pathway fuelled by *flam*, the main source of piRNAs in follicle cells, protects the germline against these deleterious elements[20,21,26]. Previous observations of *flam* were limited to species within the *melanogaster* subgroup[18,30,31,34]. Our identification of *flam* within the *suzukii* subgroup together with the absence of any piRNA cluster at the *flam*-syntenic region in and beyond the *rhopaloa* subgroup, places the emergence of *flam* between 13.3 and 15.1 MYA, long before the *melanogaster* subgroup separated from the remainder of the *melanogaster* group around 6.8 MYA.

Despite the absence of a *flam* locus in the *rhopaloa* subgroup and beyond, we identified several unistrand piRNA clusters with characteristics similar to *flam*. Across 24 *Drosophila* species, we identified five additional *flam*-like loci, including *flamlike3* and *flamlike5* present in several species across the *obscura* group. Of note, we confirmed that

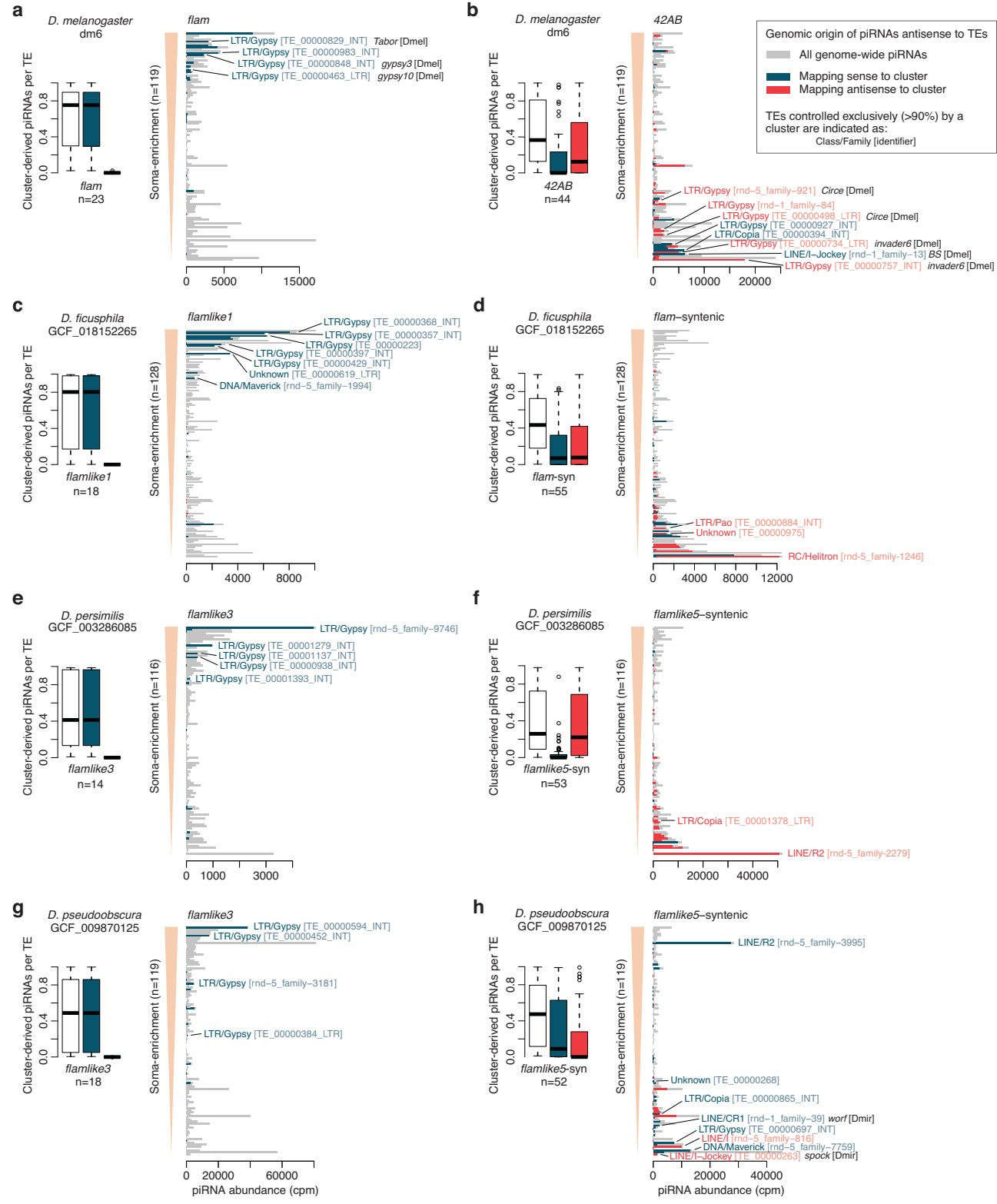

**Fig. 7 | Several *Gypsy* transposons are exclusively regulated by *flamlike1* and *flamlike3*.** Genomic origin of piRNAs that are antisense to transposons in *D. melanogaster* (**a**, **b**), *D. ficusphila* (**c**, **d**), *D. persimilis* (**e**, **f**), and *D. pseudoobscura* (**g**, **h**). Barplots (right) display the number of uniquely mappable piRNAs against each TE in soma-enriched (**a**, **c**, **e**, **g**) or whole ovary (**b**, **d**, **f**, **h**) samples. The TEs are arranged in decreasing order following their somatic-to-germline enrichment. The number of piRNAs that map to the indicated clusters are coloured according to cluster strand (sense, blue; antisense, red) and piRNAs mapping elsewhere in the genome are shown in grey. Labels are shown for subfamilies that are exclusively controlled (>90% of piRNAs) by a cluster and best hit to known TEs are indicated if available

(80/80/80 rule). Boxplots (left) summarise the fraction of piRNAs antisense to individual transposons derived from each cluster. Total cluster-derived piRNA abundance (white) are further subdivided into the sense (blue) and antisense (red) cluster strands. The number of transposon subfamilies covered by each cluster (>10 reads) are indicated under each boxplot. Boxplots show median (central line), interquartile range (IQR, box), and minimum and maximum values (whiskers, at most 1.5*IQR). Pooled counts from 2–4 biological replicates. Abbreviations: TE transposon, cpm counts per million. Source data are available in the source data file.

*flamlike3* produces abundant somatic piRNAs despite the lack of *fs(1)Yb*, a key factor for efficient *flam* processing[47,48], across the *obscura* species group (Fig. S26). This supports recent observations of mechanistic divergence, including the loss of *fs(1)Yb* or *Ago3*, across some *Drosophila* species[39]. Analysis of the cluster content revealed that all five *flam*-like loci specifically capture remnants of ERVs in the antisense orientation, consistent with strong selective pressure to generate piRNAs against these elements in somatic cells (Fig. 1). We speculate that all *Drosophila* species use *flam*-like piRNA clusters in a somatic branch of the pathway that specifically evolved to repress ERVs. While *flam*-like clusters have not been detected in all species, we have consistently identified somatic unistrand piRNA clusters in all species where we performed soma-enriched sRNA-seq. Furthermore, we did not observe any *Drosophila* species lacking the presence of *env*-containing *Gypsy*-family elements. More unistrand piRNA clusters are therefore likely to be discovered as we gain access to more sequencing data and improved genome assemblies in the future.

Interestingly, all five piRNA generating loci show substantial size, comparable to their *D. melanogaster* counterpart. Simulations also show that a single large piRNA cluster in a region without recombination is the most efficient way to stop transposon invasion[49]. In agreement with this model, individual dual-strand piRNA clusters have previously been shown to be dispensable for transposon control[50], and are often not conserved by synteny across closely related *Drosophila* species[17,50], with their content varying even amongst strains of the same species[51,52]. Of note, however, the machineries responsible for the dual-strand piRNA cluster expression and export are essential for transposon control and fertility[16,35,53–57]. Most *flam*-like unistrand clusters reported here follow the pattern of a single large locus, supporting the above model. We hypothesise that in addition to being the most efficient way of stopping TE invasion, this may enforce rigid natural selection, as disruption of *flam*-like piRNA clusters likely result in sterility. The recurring presence of unistrand clusters across the *Drosophila* genus strongly argues for an essential role of these loci, perhaps as a means to produce piRNAs in the soma without access to the germline piRNA expression and export machinery.

Surprisingly, we detected several cases of synteny between unistrand and dual-strand clusters. The *flam*-syntenic region in *D. ficusphila* harbours a dual-strand cluster and several *pseudoobscura* subgroup species have a dual-strand cluster at the *flamlike5*-syntenic location. Our data hint towards a conversion over time of unistrand loci into dual-strand clusters (Fig. 8a), or vice versa (Fig. 8b), although it could also reflect a propensity to repeatedly form piRNA clusters at specific genomic locations, likely those with low recombination rate and low selective pressure[50]. In support of the conversion model, signatures of germline *flam* expression have been observed in *D. simulans* and *D. mauritiana*[58]. Additionally, we observed germline expression towards the 3′ ends of *D. erecta* and *D. yakuba flam*-syntenic clusters. Together, this suggests that unistrand piRNA clusters can lose their somatic identity over time, particularly towards their 3′ end (Fig. 8c, d). Some dual-strand clusters thus could be vestigial, where the locus was retained, and its extant function was either acquired after or during the transformation. It will be interesting to determine whether the *D. ficusphila flam*-syntenic dual-strand cluster was initially unistranded and lost its promoter, or whether it emerged as a dual-strand cluster that gained a promoter in the *suzukii/melanogaster* ancestor.

Although their transcriptional regulation may differ, the recurrent emergence of *flam*-like loci across the *Drosophila* genus and the wider presence of unistrand clusters within the animal kingdom hints at convergent evolution, where this mechanism is best equipped to antagonise TE mobilisation. Together, our study opens the door to understanding the co-evolution between virus-like *Gypsy*-family transposons and the host defence mechanisms that silence them. Further characterisation of these novel piRNA clusters as well as the piRNA pathway machinery in these species will allow us and others to

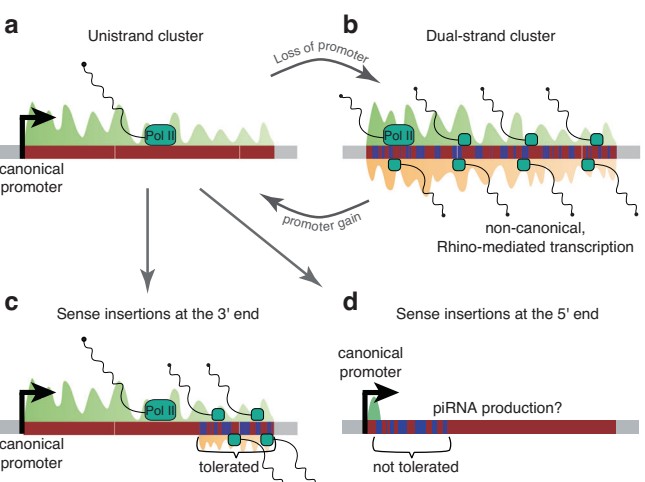

**Fig. 8 | Model of piRNA cluster conversion.** Conversion between unistrand (**a**) and dual-strand (**b**) piRNA clusters. Transposons are present either in sense (blue) or antisense (red) orientation relative to the cluster transcript(s). Produced piRNAs mapping to the sense (green) or antisense (orange) strand are shown. Once a promoter active in the soma is gained (**a**), selection will favour antisense insertions to ensure that transposon-complementary piRNAs are produced. In the absence of a promoter (**b**), the cluster can only be transcribed in germ cells, where the germline-specific branch of the piRNA pathway produces transcripts from both strands. The strand bias is therefore lost over evolutionary time. **c**, **d** Selective constraints acting on unistrand piRNA clusters. Transposon insertions in sense orientation are tolerated towards the 3′ end (**c**) but are rarely observed at the 5′ end (**d**). This may indicate that the region closer to the promoter is under stronger selective pressure. Alternatively, insertions in sense orientation may introduce polyadenylation signals causing early transcription termination, abolishing the production of essential piRNAs targeting specific TEs (**d**).

test several long-standing hypotheses regarding piRNA cluster emergence, transcriptional regulation, and the licensing of their transcripts for piRNA biogenesis.

## Methods

### Genome assemblies and nomenclature

We strived to collect as many high-quality genome assemblies as possible, including multiple ones for the same species when available, to ensure that we maximise the chance to detect novel unistrand clusters and to assess consistency across isolates. In total, we used 193 assemblies from 119 species. All downloads and processing were done by custom scripts (see "Genome_assemblies" at https://github.com/susbo/Drosophila_unistrand_clusters) and are summarised below.

We used two resources of mostly long-read assemblies representing 15 genomes for 15 species[33] and 104 genomes for 101 species[32]. The latter 104 assemblies included a re-assembly of the first 15 assemblies. Although the re-assemblies generally had higher BUSCO scores compared to the original ones[32], we kept both versions for the analysis.

Assemblies for 36 species annotated by the NCBI Eukaryotic Genome Annotation Pipeline (listed on https://www.ncbi.nlm.nih.gov/genome/annotation_euk/all) from any species within the *Drosophila* genus were downloaded on three separate occasions (2020-10-17, 2021-10-20, and 2022-04-26). Only the most recently annotated genome assembly is listed for each species and as a result, 19 species were represented by a single assembly and 15 species were represented by two different assemblies.

Additionally, we downloaded the droEre1, droSec1, droSim1, droYak2, droAna2, droPer1, dp3, droMoj2, droVir2, and droGri1 assemblies from the UCSC Genome Browser (http://hgdownload.soe.ucsc.edu/goldenPath). GCF_000754195.2 from *D. simulans* and

GCF_000005975.2 from *D. yakuba* from NCBI RefSeq and another 12 assemblies used in a recent study of *Drosophila* phylogeny[59] from the NCBI GenBank.

Genome assemblies downloaded from NCBI or the UCSC Genome Browser retained their original identifier. For assemblies downloaded from the 15 or 101 genomes resources, we used 'd15genomes' and 'd101g', respectively. For NCBI genomes, the contig/scaffold identifiers were simplified in UCSC Genome Browser-style based on information in the FASTA description lines. Unplaced contigs/scaffolds were referred to as 'chrUn_nnn' where 'nnn' refers to the numerical part of the 'NW_nnn' identifier, and contigs/scaffolds associated with a chromosome were referred to in 'chrN'_rand_nnn' format. The script used for the replacements is available in the repository above. Species abbreviations and all studied assemblies are listed in Supplementary Data 7.

## Synteny analysis for *flam* conservation
Since the clusters themselves are not conserved, we used a synteny analysis (see "Synteny_clusters" at https://github.com/susbo/Drosophila_unistrand_clusters). Briefly, we used the *D. melanogaster* genome as a reference and extracted the 20 unique up- and downstream genes, excluding tRNA, miRNA, snoRNA, asRNA, and sisRNA. For each gene, we extracted the coding sequence (protein-coding genes) or the full transcript (all others). Next, we mapped these sequences onto the genome of interest using blat (v36x6, -minIdentity=25) and filtered the results to keep the best hit (pslCDnaFilter, -minCover=0.2 -globalNearBest=0.0). Finally, we constructed a candidate list with all genomic regions that had at least two gene hits within 1 Mb. These candidate regions were then manually inspected for the presence of a transposon-rich area at the expected syntenic location. Clusters running into assembly breakpoints were labelled as either 5' or 3', depending on whether they were located next to up- or downstream genes.

## UCSC Genome Browser shots
We prepared a genome browser assembly hub covering all 193 assemblies, largely following the instructions on http://genomewiki.ucsc.edu/index.php/Assembly_Hubs. For species with NCBI gene predictions available, we used gtfToGenePred to convert the annotations from GTF to genePred format, followed by genePredToBigGenePred to convert it to bigGenePred format. Gene identifiers were replaced by gene symbols, if available, and the resulting file was converted to bigBed. Tracks displaying the best *D. melanogaster*, *D. grimshawi*, and *D. pseudoobscura* gene mappings were prepared for all genomes using blat (v36x6, -minIdentity=90) followed by pslCDnaFilter (-minCover=0.5 -globalNearBest=0.0). The resulting psl file was converted to genePred format using mrnaToGene. Gene identifiers were replaced by symbols and identical names were numbered to allow search. The edited GenePred file was converted to bigBed via bigGenePred. To construct repeat tracks (RepeatMasker, EDTA, or final TE libraries), we followed the instructions on http://genomewiki.ucsc.edu/index.php/RepeatMasker, enabling the display of the tracks, coloured by strand, and grouped by repeat type. De novo identified clusters (proTRAC) were converted from GTF to GenePred, followed by BigGenePred and BigBed. RNA-seq and sRNA-seq tracks were displayed as standard bigWig tracks produced by deepTools. All genome browser shots shown in this study were made by exporting the assembly hub display as a pdf, followed by manual refinement to enhance readability.

## RepeatMasker tracks
Repeat annotations were done using RepeatMasker (v4.1.2, -s -species Drosophilidae -xsmall) with the Dfam v3.5 and RepBase (RepeatMaskerEdition-20181026) databases[60].

## Mappability tracks
Mappability tracks for the sRNA-seq were constructed by generating all possible 26-mers from each genome (bedtools, v2.26.0), aligning them back to the genome (bowtie, v1.2.3, -S -n 2 -M 1 --best --strata --nomaqround --chunkmbs 1024 --no-unal), and converting the alignments to bigWig using deepTools bamCoverage (v3.3.2, --binSize 1 --normalizeUsing None --scaleFactor 0.038461). This will construct a per-nucleotide signal between 0 and 1, representing the ability to uniquely map reads to each position.

## LTR strand bias and transposon classes in repeat tracks
To retain consistency across all genome browser shots, the percentage of LTR transposons per strand (displayed in Fig. S1 and others) is always quantified based on the EDTA repeat annotations (see "De novo transposon annotations using EDTA"), whereas cluster content (bar graphs in Fig. S1 and others) is based on the curated transposon libraries (see "Construction of curated de novo transposon libraries" below).

## Synteny visualisation
MCScan tool[61] was used for synteny comparisons and visualisations. The input to MCScan (Python version) was gene sequences in FASTA format and coordinates in BED-like format. This tool calls LAST[62] to perform pairwise synteny search and a single linkage clustering is performed on the LAST output to cluster anchors into synteny blocks. Following this calculation, different visualisations can be produced with MCScan and hence we performed "Macrosynteny" and "Microsynteny" visualisation. The former is a karyotype plot and highlights syntenic regions across species' genomes whereas the latter offers the advantage to investigate local synteny, which focuses on gene-level. Based on the MCScan tool, we created a set of Python scripts that can be used to investigate and visualise gene synteny between *Drosophila* species. The code and examples with instructions are available on GitHub (https://github.com/marianna-trapotsi/MCScan_plot).

## Phylogenetic trees
To display our species onto a phylogenetic tree, we used a previously reported IQ-TREE maximum-likelihood analysis for 704 Drosophilidae species[63]. The phylogeny was imported using the R module treeio (v.1.10.0) and species not included in our study were dropped. Visualisation of the tree and metadata was done using ggtree (v2.0.4) and ggnewscale (v0.4.6). Time of divergence estimates were taken from another recent study of *Drosophila* phylogeny[59] based on fossil evidence.

## De novo transposon annotations using EDTA
An initial de-novo transposon library was built using EDTA (v1.9.3, --sensitive 1 --anno 1 --evaluate 1)[37]. The EDTA pipeline consists of three steps, detection of LTRs, Helitrons and TIRs. Most genomes were successfully processed, with transposons of all types being detected. However, some runs failed when one of the types were missing and we manually resumed EDTA at the next type for these genomes. Three genome assemblies (Dneo-d101g, Dsal-d101g, and Zind-d101g_BS02) that failed to run with EDTA v1.9.3 did run successfully with v1.9.6, whereas four assemblies (Daca-d101g, Dari-GCF_001654025, Dnav-GCF_001654015, Dwas-d101g) that had problems with v1.9.3 still had to be resumed with v1.9.6 due to not detecting any LTR transposons.

## De novo detection of unistrand *flam*-like clusters
To search for *flam*-like clusters, we developed a search strategy based on the known enrichment of LTR transposons arranged in the same orientation in *flam* (see "De-novo_clusters" at https://github.com/susbo/Drosophila_unistrand_clusters). Briefly, repeat annotations from the EDTA were used. The repeats were separated based on strand retaining either only LTR transposons, or all transposons with a predicted class (i.e., not unknown). Overlapping annotations were combined (bedtools merge) and strand-specific transposon coverage was computed (bedtools coverage) across the genome using a 100 kb

sliding window with a 5 kb step size (bedtools makewindows, -w 100000 -s 5000). Each genome was manually inspected for regions enriched in LTR transposons, with a strong strand bias, and located outside of centromeric or telomeric regions. This analysis was strongly contingent on assembly quality, but we nevertheless identified 15 clusters that fulfilled the outlined criteria, including several corresponding to *flam* across the *D. melanogaster* group. Six of the initial candidates were found outside of the *D. melanogaster* subgroup. Of these, two species had publicly available sRNA-seq data, and both produced large amounts of piRNAs from one strand only. We therefore concluded that the approach was working. Synteny analysis using these five clusters as starting points (described below) revealed that *D. persimilis* and *D. pseudoobscura* were syntenic.

### Fly husbandry
All *Drosophila* species were maintained at room temperature. The origin of each species and their food requirements are indicated in Supplementary Data 8.

### Small RNA-seq library preparation
Small RNAs were isolated from 16 species (2–3 replicates each) using the TraPR Small RNA Isolation Kit (Lexogen; catalogue nr. 128.24) following the manufacturer's instructions. sRNA libraries were generated using the Small RNA-Seq Library Prep Kit (Lexogen; catalogue nr. 052.96) with minor modifications. Both primers A3 and A5 as well as the primer RTP were used at 0.5x. Library size distribution was analysed on an Agilent TapeStation system using a High Sensitivity D1000 ScreenTape (Agilent Technologies; catalogue nr. 5067-5584) with High Sensitivity D1000 Reagents (Agilent Technologies; catalogue nr. 5067-5585). Libraries were pooled in equal molar ratio, quantified with KAPA Library Quantification Kit for Illumina (Kapa Biosystems; catalogue nr. KK4873) and were sequenced 50 nt paired-end on an Illumina NovaSeq 6000 or 75 nt single-end on an Illumina MiSeq sequencing platform generating 33 (±20) million reads per library.

### Soma-enriched small RNA library preparation
The soma-enrichment sRNA-seq libraries were generated for 13 species (2 replicates each) similar to published protocols[30,64], with modifications. In brief, 75-100 ovary pairs were dissected in ice-cold PBS. Ovaries were dissociated for 18 min in 0.25% Trypsin (Sigma-Aldrich; catalogue nr. T1426) at 25 °C, shaking at 800 rpm. Dissociated tissue was pushed through a 40 µm nylon mesh (Greiner Bio-One; catalogue nr. 542040) washed with equal volume Schneider 2 medium (Thermo Fisher Scientific; catalogue nr. R69007) and then pelleted. Pelleted cells were directly used as input for sRNA isolation using the TraPR Small RNA Isolation Kit (Lexogen; catalogue nr. 128.24), following the manufacturer's instructions. sRNA libraries were generated using the Small RNA-Seq Library Prep Kit (Lexogen; catalogue nr. 052.96) with minor modifications. Both primers A3 and A5 as well as the primer RTP were used at 0.5x. Library size distribution was analysed on an Agilent TapeStation system using a High Sensitivity D1000 ScreenTape (Agilent Technologies; catalogue nr. 5067-5584) with High Sensitivity D1000 Reagents (Agilent Technologies; catalogue nr. 5067-5585). Libraries were pooled in equal molar ratio, quantified with KAPA Library Quantification Kit for Illumina (Kapa Biosystems; catalogue nr. KK4873) and were sequenced 50 nt paired-end on an Illumina NovaSeq 6000 sequencing platform generating 43 (±25) million reads per library.

### Publicly available sRNA-seq data
We searched the Sequencing Read Archive (SRA) and Gene Expression Omnibus (GEO) for any sRNA-seq data from *Drosophila* species other than *D. melanogaster*. After excluding two SOLiD sequencing samples, we found 67 samples from 12 species[30,65–69], representing embryo ($n = 16$), female body ($n = 11$), female germline ($n = 2$), female soma ($n = 2$), head ($n = 18$), male body ($n = 9$), follicle-cell enriched ovary ($n = 3$), and testis ($n = 6$).

### Processing of sRNA-seq data
All sRNA-seq data were processed using the same analysis pipeline. Trim Galore! (v0.6.4, --stringency 30 -e 0.1 -a TGCTTGGACTACA-TATGGTTGAGGGTTGTA --length 18 -q 0) was first run to remove an abundant rRNA sequence, followed by a second run (--stringency 5 -e 0.1 --length 18 --max_length 35 -q 0) to remove adapter sequences (specified using 'a'), and any flanking random nucleotides ('--clip_R1' and/or '--three_prime_clip_R1' with appropriate arguments). All samples and their adapter sequences are listed in Supplementary Data 9.

The processed reads were mapped to a miRNA hairpin database (miRBase release 22.1)[70] using bowtie (v1.2.3, -S -n 2 -M 1 -p 20 --best --strata --nomaqround --chunkmbs 1024) with '--un' and '--max' to extract unmapped reads. Reads not mapping to miRNAs were aligned to the respective reference genomes using bowtie (-S -n 2 -M 1 -p 20 --best --strata --nomaqround --chunkmbs 1024). Multi-mapping reads were extracted into a separate BAM file using awk (MQ < 10). For the cluster content analysis (Fig. 7), the trimmed and filtered reads were also aligned to curated transposon libraries using bowtie (v1.2.3, -S -n 2 -M 1 -p 20 --best --strata --nomaqround --chunkmbs 1024). Alignment metrics are available in Supplementary Data 9.

The BAM files were converted to bigWig using bamCoverage from deepTools[71] (v3.3.2, --binSize 1 --ignoreForNormalization chrM --normalizeUsing CPM --exactScaling --skipNonCoveredRegions --minFragmentLength 23 --maxFragmentLength 30) and additionally '--filterRNAstrand' to separate the two strands, '--scaleFactor' to scale counts per million to reflect all mapped reads, and optionally '--minMappingQuality 50' when extracting uniquely mapped reads.

### RNA-seq library preparation
The RNA-seq libraries were generated for 15 species (2–4 replicates each). Briefly, ovaries from 10-20 flies were dissected in ice-cold PBS and total RNA was extracted using TRIzol (Thermo Fisher Scientific; catalogue nr. 15596026), following the manufacturer's instructions. Ribosomal RNA was depleted using RiboPOOL (siTOOLs Biotech; catalogue nr. dp-K024-000007) following the manufacturer's protocol. Additionally, we re-sequenced (paired-end) four published *D. melanogaster* samples treated with RiboZero as previously described[72]. RNA-seq libraries were produced using NEBNext Ultra Directional Library Prep Kit for Illumina (New England BioLabs; catalogue nr. E7420L), following the manufacturer's instructions for rRNA depleted RNA. Library size distribution was analysed on an Agilent TapeStation system using a High Sensitivity D1000 ScreenTape (Agilent Technologies; catalogue nr. 5067-5584) with High Sensitivity D1000 Reagents (Agilent Technologies; catalogue nr. 5067-5585). Libraries were pooled in equal molar ratio, quantified with KAPA Library Quantification Kit for Illumina (Kapa Biosystems; catalogue nr. KK4873) and sequenced paired-end 50 nt on an Illumina NovaSeq 6000 generating 25 (±11) million reads per library.

### Soma-enriched RNA-seq library preparation
The soma-enrichment RNA-seq libraries were generated for 5 species (2 replicates each). Enrichment for somatic cells was done identically as described for the soma-enriched sRNA-seq libraries, except that 35-50 ovary pairs were used as starting material. Pelleted cells were directly used as input for RNA isolation using the TRIzol (Thermo Fisher Scientific; catalogue nr. 15596026). RNA was treated with DNase (New England BioLabs; catalogue nr. M0303) followed by ribosomal RNA depletion using RiboPOOL (siTOOLs Biotech; catalogue nr. dp-K024-000007) following the manufacturer's protocol. RNA-seq libraries were produced using NEBNext Ultra Directional Library Prep Kit for Illumina (New England BioLabs; catalogue nr. E7420L), following the manufacturer's instructions for rRNA depleted RNA. Library size distribution was analysed on an Agilent TapeStation system using

a High Sensitivity D1000 ScreenTape (Agilent Technologies; catalogue nr. 5067-5584) with High Sensitivity D1000 Reagents (Agilent Technologies; catalogue nr. 5067-5585). Libraries were pooled in equal molar ratio, quantified with KAPA Library Quantification Kit for Illumina (Kapa Biosystems; catalogue nr. KK4873) and sequenced paired-end 50 nt on an Illumina NovaSeq 6000 generating 42 (±7.1) million reads per library.

## Publicly available RNA-seq data

We downloaded modENCODE RNA-seq data from *Drosophila* species other than *D. melanogaster* and additional *D. innubila* samples[73,74]. We included 58 samples from 8 species, representing embryo ($n = 16$), female body ($n = 18$), female head ($n = 1$), larvae ($n = 2$), male body ($n = 18$), male head ($n = 1$), and pupae ($n = 2$).

## Processing of RNA-seq data

Trim Galore! (v0.6.4, --stringency 6 -e 0.1) was used to remove adapters and low-quality bases. Alignment was done using HiSeq2[75] (v2.2.0, -max-seeds 100 -q -k 1), keeping at most one alignment for each read. Multi-mapping reads were extracted into a separate BAM file using awk (MQ < 10). The BAM files were converted to bigWig using bamCoverage from deepTools[71] (v3.3.2, --binSize 1 --ignoreForNormalization chrM --normalizeUsing CPM --exactScaling --skipNonCoveredRegions) and additionally '--filterRNAstrand' to separate the two strands, '--scaleFactor' to scale the counts per million to all mapped reads, and optionally '--minMappingQuality 50' when extracting uniquely mapped reads. Alignment metrics are available in Supplementary Data 10.

## Ping-pong and phasing analyses

Ping-pong and phasing analyses were performed for reads of length 24-28 nt mapping to each cluster region. The deepTools module bamCoverage was used to extract the number of 3′ and 5′ ends mapping to each position and strand. Ping-pong and phasing signature was calculated following the strategy in[76]. In short, we calculated the ping-pong signature using a 5′ end overlap score for overlap $x$ nt as

$$s_x = \sum_{i \in all\ positions} n_i m_{i+x} \tag{1}$$

where $n_i$ is the number of 5′ ends mapping at the plus strand position $i$ and $m_{i+x}$ is the number of 5′ ends mapping at the minus strand position $i+x$. The fraction of overlapping reads involved in ping-pong was calculated as $s_{10}/(s_1 + \ldots + s_{20})$. A z10 score was defined as $(s_{10}-mean(s_1,\ldots,s_9,s_{11},\ldots,s_{20}))/stdev(s_1,\ldots,s_9,s_{11},\ldots,s_{20})$.

For the phasing signature, we calculated a 3′ to 5′ end score for distance $y$ as

$$h_y = \sum_{i \in all\ positions} \min(n_i, m_{i+y}) \tag{2}$$

where $n_i$ is the number of 3′ ends mapping at position $i$ and $m_{i+y}$ is the number of 5′ ends mapping at position $i+y$ at the same strand. The fraction of closely mapped reads with phasing signature was calculated as $h_1/(h_1 + \ldots + h_{20})$. A z1 score was calculated as $(h_1-mean(h_2,\ldots,h_{20}))/stdev(h_2,\ldots,h_{20})$. Phasing calculations were done for the plus and minus strand separately.

## Detection of regions syntenic to *flam*-like clusters

Synteny analysis for *flam*-like clusters was performed using the same strategy as for *flam*, except that Augustus gene predictions (v3.3.2, --species=fly --UTR=off --singlestrand=true) were used instead of Fly-Base annotations. The MAKER-masked genome from the EDTA output was used as genome input to Augustus. Full transcript and coding sequences were extracted from the annotations. Sequences with strong hits to the raw transposon libraries were excluded (blat, -q=dna -t=dna -minIdentity=25; pslCDnaFilter, -minCover=0.2 -globalNearBest=0) and gene predictions shorter than 200 nt were excluded. The blat identity threshold was reduced to 20.

Additionally, within the *obscura* group (*flamlike5*), the closest flanking genes displayed good conservation and we used these to search for syntenic regions using our UCSC Genome Browser session.

## ATAC-seq library preparation

ATAC-seq was performed for nine species similar as described in[77]. Briefly, 6-12 ovary pairs of yeast-fed flies were dissected in ice-cold PBS and centrifuged for 5 min at 500 g at 4 °C. Ovaries were lysed in Resuspension Buffer (RSB, 10 mM Tris-HCL pH 7.4, 10 mM NaCl, 3 mM MgCl2 in nuclease free water) containing 0.1% NP40, 0.1% Tween20, and 0.01% Digitonin and washed out with cold RSB containing 0.1% Tween-20. The transposition reaction was performed with 0.33x PBS, 0.01% digitonin, 0.1% Tween-20, 1x TD buffer and 100 nM transposase (Illumina Tagment DNA Enzyme and Buffer Small Kit; catalogue nr. 20034197). Samples were incubated for 1 h at 37 °C in a thermomixer mixing at 1000 rpm. The transposed fragments were isolated using the DNA Clean and Concentrator-5 Kit (Zymo Research; catalogue nr. D4014). Library was PCR amplified for 5 cycles using the NEBNext High-Fidelity MasterMix (New England BioLabs; catalogue nr. M0541S) followed by qPCR amplification to determine the exact number of additional cycles required for optimal library amplification. Amplified DNA library was purified using the DNA Clean and Concentrator-5 Kit (Zymo Research; catalogue nr. D4014) and further cleaned using AMPure XP beads (Beckman Coulter; catalogue nr. A63881). 100-600 bp fragments were selected on a 2% agarose gel cassette using the Blue Pippin (Sage Science; catalogue nr. NC1025035). Library size distribution was analysed on an Agilent 2100 bioanalyzer using the High Sensitivity DNA Kit (Agilent Technologies; catalogue nr. NC1738319). Libraries were pooled in equal molar ratio, quantified with KAPA Library Quantification Kit for Illumina (Kapa Biosystems; catalogue nr. KK4873) and were sequenced 50 nt paired-end on an Illumina NovaSeq 6000 platform or Illumina MiSeq sequencing platform generating 4.5-25.6 million paired-end reads per library.

## Processing of ATAC-seq data

The quality of raw reads was assessed using FastQC (v0.11.8). Cutadapt (v1.18, default parameters) was used to trim Nextera Transposase Adapters from the paired-end reads. The trimmed and paired reads were aligned to the respective genome assembly using Burrows-Wheeler Aligner (BWA) (v0.7.17, bwa mem -M -t 4)[78]. Picard tool (v2.9.0) was used to mark duplicates. SAMtools (v1.9) was used for indexing and filtering. Quality metrics for the aligned ATAC-seq reads were assessed using ataqv (v1.0.0) (https://github.com/ParkerLab/ataqv)[79]. ATAC-seq peaks were called with MACS2 (v2.1.1, --nomodel --shift −37 --extsize 73 -g dm --keep-dup all -q 0.05)[80]. The bamCoverage module from deepTools (v3.5.1, --binSize 1 --normalizeUsing RPKM --effectiveGenomeSize 125464728) was used to generate normalized bigWig files. Peak intersections were performed using bedtools (v2.30.0)[81]. Conservation of ATAC-seq peaks were assessed by performing LiftOver (UCSC) or NCBI BLAST+ (2.14.0 release) of the ATAC-seq peak regions to find orthologous genomic regions in the other species and checking if they also have an ATAC-seq peak in that region. Genome browser visualizations were done using the UCSC Genome Browser. Alignment metrics are available in Supplementary Data 11.

## Prediction of major piRNA clusters

Cluster predictions were performed using proTRAC (v2.4.4, -pdens 0.01 -swincr 100 -swsize 1000 -clsize 5000 -1Tor10A 0.75 -clstrand 0.5 -pimin 23 -pimax 30 -pisize 0.75 -distr 1-99 -nomotif -format SAM)[40] using all available sRNA-seq libraries except head, female_soma,

male_body and OSC, with '-repeatmasker' set to RepeatMasker annotations generated by EDTA, and with '-geneset' set to NCBI gene predictions, if available.

Clusters within 40 kb from each other were combined for the analyses. Genes flanking the major *D. melanogaster* clusters were mapped onto each genome using BLAT (v36x6, -minIdentity=25), filtered to retain only the best hit (pslCDnaFilter, -minCover=0.2 -globalNearBest=0.0) and the predicted clusters were subsequently annotated by how many of these genes that were within 1 Mb.

Cluster predictions used in the soma-enrichment analysis were performed using the same strategy but restricted to libraries generated for this study and using either only soma-enriched or only total sRNA-seq libraries (2–3 replicates per species and library type). Clusters identified using either somatic or total libraries were concatenated and any clusters within 40 kb from each other were merged. Clusters of size <35 kb were discarded to enable analysis of strand biases across major clusters. Total piRNA coverage per cluster was normalised to counts per million and calculated for soma-enriched and total libraries separately. Somatic clusters were defined as clusters with at least 2-fold soma enrichment over total libraries, and germline clusters were defined as being higher expressed in the total ovary libraries.

### Genome assembly QC

To assess genome assembly quality, we calculated number of sequences (NN) and estimated genome contiguity using N50 (Supplementary Data 7). NN and N50 values were obtained using calN50 (RRID:SCR_022015, https://github.com/lh3/calN50). A genome assembly was considered to have adequate quality if N50 > 1,000,000 and NN < 3,000. Species with only low-quality assemblies are indicated in Fig. 5g.

### Construction of curated de novo transposon libraries

In addition to the consensus sequences obtained from EDTA, we also used the RepeatModeler (v2.0.1) output within the EDTA folders to improve detection of LINE elements. We reasoned that we could not provide a list of known LINEs to EDTA, since that would mainly reflect *melanogaster* transposons and would bias the comparisons between the *melanogaster* subgroup and other species.

EDTA and RepeatModeler consensus sequences were combined and further processed using a custom pipeline (see "Transposon_libraries" at https://github.com/susbo/Drosophila_unistrand_clusters). Briefly, raw sequences classified as rRNA, snRNA, tRNA, ARTEFACT, or Simple_repeat were removed, and remaining sequences were clustered using cd-hit-est (v4.8.1, -G 0 -g 1 -c 0.90 -aS 0.90 -n 8 -d 0 -b 500) to combine any sequences with ≥90% identity across ≥90% of the length. Custom scripts were used to select one representative sequence from each cluster, maximising both the number of high-quality genomic hits (blastn, filtered to cover at least 50% of the query sequence) and the length of the sequence. Sequences with fewer than 2 high-quality genomic hits were removed from the transposon libraries.

To prioritise sequences and to detect known transposon domains, we mapped all consensus sequences to *env*, *gag* and *pol* ORFs from RepeatPeps.lib in RepeatMasker (v4.1.2) using blastx (v2.10.0, -max_target_seqs 100 -evalue 1e-3). Sequences covering at least 50% of the full peptide domain were considered true hits.

To detect previously described subfamilies, we used the *Drosophila* transposon canonical sequences database (v10.2, https://github.com/bergmanlab/transposons), including 180 consensus sequences from seven species. We used full_blast (https://github.com/rimjhimroy/Transposon80-80-80, -p megablast -i 95 -qc 98 -l 80 -t 10) to identify hits with ≥95% identity, hit length ≥80 nt, and ≥98% query coverage (i.e., "95/80/98 rule"[82,83]) to the canonical sequences, which were considered to belong to the same subfamily. This was repeated using 90/80/90 and 80/80/80 thresholds to detect more distant similarities.

Transposon class and family were predicted using the Repeat-Classifier script in RepeatModeler (v2.0.2a), configured to use RepeatMasker (v4.1.2) with the Dfam (v3.5) and RepBase (RepeatMaskerEdition-20181026) databases[60].

The curated and annotated transposon consensus sequences have been made available (https://github.com/susbo/Drosophila_TE_libraries).

### Mapping of sRNA-seq and RNA-seq across species

To evaluate whether different *Drosophila* species express similar piRNAs (Fig. S23), we extracted 100,000 filtered reads from each sRNA-seq library and aligned them across all 193 assemblies using bowtie (v1.2.3, -S -n 2 -M 1 -p 10 --best --strata --nomaqround --chunkmbs 1024 --no-unal). Alignment rate for multiple assemblies of the same species were averaged. Most libraries displayed near-zero alignment rate across all species except very closely related ones. A small number of libraries showed elevated baseline alignment, likely due to the presence of other small RNAs (rRNA, tRNA) with high conservation. To remove this effect, we subtracted the median alignment rate from each library, which roughly corresponds to removing reads mapping to all species. A similar strategy was used to map RNA-seq libraries across species (Fig. S24) using 200,000 reads aligned with hisat2 (v2.2.0, --max-seeds 100 -q -k 1 -p 10 --no-unal --new-summary --summary-file). Similarly, the median alignment rate was subtracted from each library to reduce false hits driven by abundant non-coding RNAs.

### Cluster content analyses

Cluster content analyses in Fig. 6 uses the curated transposon libraries either in the form of relative transposon coverage (Fig. c-e), absolute transposon coverage (Fig. 6f) or the number of transposon subfamilies (Fig. 6g). To obtain these estimates we ran RepeatMasker (v4.2.1, -s -xsmall) with the curated transposon libraries (-lib), followed by a modified version of extractNestedRepeats.pl (from UCSC Genome Browser) that returned all repeats (not only the nested ones) as a BED file, which was further analysed to ensure that each base was only counted once in case of any overlapping annotations. All unistrand clusters were predominantly occupied by transposons on their antisense strand, and to be able to compare transposon occupancy to dual-strand clusters, we assigned a strand to each dual-strand clusters as the strand opposite to most transposons. For the remaining analyses, we used buildSummary.pl (part of RepeatMasker), to obtain genome-wide and within transposon estimates of transposon coverage (Fig. 6f) and copy number (Fig. 6g).

To determine whether clusters were more likely to have captured *Gypsy*-family LTR transposons compared with other LTR transposons we defined a Gypsy enrichment ratio as

$$\text{Gypsy enrichment} = \frac{P(x \in \text{Gypsy} | x \in \text{LTR}, x \in \text{Captured})}{P(x \in \text{Gypsy} | x \in \text{LTR}, x \in \text{Not captured})} \quad (3)$$

And similarly, we defined an *env* enrichment ratio as

$$\text{env enrichment} = \frac{P(x \in \text{env} | x \in \text{Gypsy}, x \in \text{Captured})}{P(x \in \text{env} | x \in \text{Gypsy}, x \in \text{Not captured})} \quad (4)$$

Where $x$ is a transposon subfamily, Gypsy is the set of all *Gypsy*-family transposon subfamilies, LTR is the set of all LTR transposons, Captured is the set of all transposon subfamilies found inside the cluster region and Not captured is the set of all other transposon subfamilies.

### Cluster expression analysis

Reads mapping uniquely to each genome were intersected with cluster coordinates using bedtools intersect. Resulting counts were normalised to the total number of reads mapping to each genome.

## Determination of regulatory potential of individual clusters

For the cluster content analysis (Fig. 7), we considered only reads of length 24-28 nt that mapped uniquely to the curated transposon libraries. For each species, a set of 100 piRNA-regulated transposons were defined for whole ovary and soma-enriched ovary, separately, by ranking the sequences in the curated transposon library by the number of piRNAs mapping to them across all replicates. The rankings were highly similar between whole ovary and soma-enriched ovary, and in total 116 to 128 transposon subfamilies were selected per species. To enable comparison of the counts in soma-enriched and whole ovary libraries, we derived cpm values by normalising the counts to the total number of reads mapping to the genome. Soma-enrichment per transposon subfamily was calculated as the difference in cpm between the pooled soma-enriched and whole ovary libraries.

Next, we further restricted the analysis to reads that also mapped uniquely to the genome assembly and used the read identifiers to assigned transposon identity and transposon strand to each genome-mapping read. Finally, the reads were intersected to piRNA cluster coordinates (bedtools intersect, v2.26.0) in strand-specific mode to allow determination of whether the reads originated from the sense or antisense strand of a cluster. For the intersection analysis, dual-strand clusters were assumed to be located on the + strand. Metadata for the curated transposon libraries were obtained as described previously under "Construction of curated de novo transposon libraries".

## Conservation of piRNA pathway genes

To avoid false hits to conserved protein domains and to increase sensitivity compared with sequence-based searches, we employed a synteny-based search strategy (see "Synteny_biogenesis_genes" at https://github.com/susbo/Drosophila_unistrand_clusters). Briefly, we used the *D. melanogaster* genome as a reference and extracted the 20 closest genes up- and downstream genes, excluding tRNA, miRNA, snoRNA, asRNA, and sisRNA. For each gene, we extracted the coding sequence (protein-coding genes) or the full transcript (all others). Next, we mapped these sequences onto the genome of interest using blat (v36x6, -maxIntron=500000 -minMatch=2 -minScore=30 -oneOff=1 -minIdentity=10) and filtered the results to keep the best hit (pslCDnaFilter, -minCover=0.1 -globalNearBest=0.0). Finally, we constructed a hit list with all genomic regions that had at least two hits within 1 Mb from another. Typically, this resulted in a single hit, which was manually inspected for the presence of the gene of interest.

## Reporting summary

Further information on research design is available in the Nature Portfolio Reporting Summary linked to this article.

## Data availability

The high-throughput sequencing data and genome browser tracks generated in this study have been deposited at GEO under accession code GSE225889. The transposon libraries and metadata generated in this study are available on GitHub through https://github.com/susbo/Drosophila_TE_libraries[84]. Source data are provided with this paper.

## Code availability

Custom code is available at GitHub (https://github.com/susbo/Drosophila_unistrand_clusters)[85].

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

## Acknowledgements

We thank Hannon group members for fruitful discussions, Holly Goodrick for help with library preparation and Emma Kneuss for providing early access to unpublished sRNA-seq libraries. We thank the Scientific Computing core at the CRUK Cambridge Institute for HPC resources and the Genomics core for sequencing services. GJH is a Royal Society Wolfson Research Professor (RSRP\R\200001). This research was funded in whole, or in part, by Cancer Research UK (G101107) and the Wellcome Trust (110161/Z/15/Z). The following species were kindly provided by the indicated laboratories: *D. mojavensis, D. persimilis* and *D. pseudoobscura* from Ben Longdon, University of Exeter, UK; *D. ficusphila* from Geoffrey Findlay, College of the Holy cross, Worcester, MA, USA; *D. simulans* from David Stern, Janelia Research Campus, VA, USA; *D. erecta, D. yakuba, D. ananassae* and *D. virilis* from Simon Collier at the Department of Genetics, Fly Facility, University of Cambridge, UK.

## Author contributions

J.v.L., G.J.H., S.B. and B.C.N. conceived the study. S.B., J.v.L. and B.C.N. designed the experiments and interpreted the results. A.A. performed ATAC-seq experiments and analysis. J.v.L. performed all other wet-lab work and preliminary computational analysis. S.B. performed all other computational work, except the MCScan analysis that was performed by MAT. B.C.N., G.J.H. and S.B. supervised the work. J.v.L, S.B. and B.C.N. wrote the manuscript with input from GJH. All authors read and approved the final version.

## Competing interests

The authors declare no competing interests.
