## [Peer Review File · Nature Communications]

Reviewers' Comments:

Reviewer #1:

Remarks to the Author:

This manuscript focuses on an emerging topic—the evolution of piRNA clusters. The authors combined bioinformatics tools and deep sequencing to identify and characterize flam and flam-like unistrand piRNA clusters across *Drosophila* species. The flam cluster is best studied in *Drosophila melanogaster* and it plays an essential role in suppressing Gypsy transposon family in somatic follicle cells. The authors provided evidence for the presence of flam clusters in *melanogaster*, *suzukii*, and related subgroups. In addition, they discovered a set of flam-like genomic loci in some *Drosophila* species. Using synteny analysis, they found dual-strand piRNA cluster in one species can be syntenic with flam unistrand cluster, indicating possible conversion between dual and unistrand clusters. Overall, their findings raise some interesting questions on piRNA cluster evolution and transposon silencing. While I found the study informative and well-written, I have some major concerns that should be addressed before the manuscript can be accepted for publication. Specifically, the study is descriptive, and to provide mechanistic insights, the authors need to address the following questions:

Major Comments

1. Are piRNAs derived from flam-like clusters required to suppress ERV? While the bioinformatic analysis is suggestive, the authors need to provide experimental data to confirm the conserved function of flam/flam-like clusters.
2. What is the mechanism of conversion between unistrand and dual-strand piRNA clusters? Authors should consider to conduct genetic manipulation to force the conversion in one species.
3. In Fig. 5g, is absence/presence of flam/flam-like clusters correlate with the quality and completeness of genomes?

Minor Comments and Errors that should be addressed

1. Fig. 4f was discussed prior to Fig. 4e.
2. In the legend of figure 5, a mix of uppercase and lowercase letters was used.
3. In Fig. 7, the direction of loss/gain of promoter is wrong.

Reviewer #2:

Remarks to the Author:

The manuscript by van Lopik et al presents a thorough study of flam and flam-like clusters in over a hundred of *Drosophila* species. This work advances our knowledge of genomic origins of gypsy-targeting piRNAs outside *D. melanogaster*. Data presented here help in understanding the evolution of piRNA transposon defense in the somatic cells of fly ovaries. I support the publication of this work in *Nature Communications* after the two suggestions listed below are incorporated in the revised manuscript.

-In the discussion section, when comparing the evolutionary turnover rates of germline and flam/flamlike clusters, authors conclude that flam/flamlike clusters are more conserved. A quantitative assessment would be useful here and should serve as a more compelling evidence.

-Presence of a promoter driving somatic expression of a locus is hypothesized by authors as a possible route to establishing a unistrand cluster. Having identified flam and flamlike clusters in so many species, authors could attempt at determining such promoter sequences, maybe even the unifying features of these regulatory elements.

Response to reviewers:

We thank both reviewers for their overall positive feedback and hope that we have addressed their comments in the updated manuscript and point-to-point response below.

Furthermore, we have also added the following new data:

- 11 libraries of ribo-depleted RNA-seq from 6 species (*D. azteca*, *D. biarmipes*, *D. bifasciata*, *D. ficusphila*, *D. suzukii*, *D. takahashii*)
- 10 libraries of somatic-enrichment RNA-seq from 5 species (*D. biarmipes*, *D. ficusphila*, *D. melanogaster*, *D. persimilis*, *D. pseudoobscura*)
- 18 libraries of ATAC-seq from ovaries from 9 species (*D. biarmipes*, *D. erecta*, *D. ficusphila*, *D. melanogaster*, *D. persimilis*, *D. pseudoobscura*, *D. similans*, *D. suzukii*, *D. yakuba*)
- 2 libraries of small RNA-seq from *D. bifasciata*
- 2 libraries of somatic-enrichment small RNA-seq from *D. suzukii*

This has allowed us to expand existing analyses including strengthening the link between *flam*-like clusters and Gypsy-family transposon repression, to define the putative promoter regions of unistrand clusters, and to confirm the somatic expression profile of *flam*-syntenic and *flam*-like clusters for one additional species on piRNA level, and now also five species on RNA-seq level. Together we believe these additions, greatly improve the manuscript and expand the conclusions we draw compared to the previous manuscript version.

Reviewer #1 (Remarks to the Author):

This manuscript focuses on an emerging topic—the evolution of piRNA clusters. The authors combined bioinformatics tools and deep sequencing to identify and characterize *flam* and *flam*-like unistrand piRNA clusters across *Drosophila* species. The *flam* cluster is best studied in *Drosophila melanogaster* and it plays an essential role in suppressing Gypsy transposon family in somatic follicle cells. The authors provided evidence for the presence of *flam* clusters in *melanogaster*, *suzukii*, and related subgroups. In addition, they discovered a set of *flam*-like genomic loci in some *Drosophila* species. Using synteny analysis, they found dual-strand piRNA cluster in one species can be syntenic with *flam* unistrand cluster, indicating possible conversion between dual and unistrand clusters. Overall, their findings raise some interesting questions on piRNA cluster evolution and transposon silencing. While I found the study informative and well-written, I have some major concerns that should be addressed before the manuscript can be accepted for publication. Specifically, the study is descriptive, and to provide mechanistic insights, the authors need to address the following questions:

Major Comments

1. Are piRNAs derived from *flam*-like clusters required to suppress ERV? While the bioinformatic analysis is suggestive, the authors need to provide experimental data to confirm the conserved function of *flam*/*flam*-like clusters.

We agree that ERV upregulation following disruption of *flam*(/-like) loci would provide additional evidence. However, introducing mutations, especially alterations that result in sterility, in non-model organisms such as other *Drosophila* species is technically challenging, if not impossible

due to the repeat content in the target regions. Considering that such piRNA cluster mutants would take at least half a year to establish, if at all possible, we argue that providing functional insight through direct cluster manipulation is beyond the scope of the current study.

Nevertheless, in our opinion our manuscript already presents a clear model for ERV suppression through soma-expressed unistrand piRNA clusters that is supported through several independent experiments and analyses. First, our soma-enriched piRNA profiling has shown that these clusters indeed produce highly abundant piRNAs in somatic cells that are capable of targeting transposons (i.e., having base-pair complementarity to transposons). Second, in agreement with this role, we observe an enrichment of ERV elements in these clusters, both when comparing to other LTR elements, or to other Gypsy elements. Third, we show that these loci are evolutionarily conserved.

To provide additional support, we have now:

- Performed additional analyses of dual-strand germline clusters as controls (**Fig. 6d,e**), showing that they lack these signatures.
- Performed new analyses to identify individual transposons subfamilies that are controlled by *flamlike1* and *flamlike3*, and dual-strand control clusters, and shown that only the former produce piRNAs specifically against soma-expressed LTR/Gypsy subfamilies, consistent with a *flam*-like role (**Fig. 7**).

In summary, we believe that our study provides compelling evidence that somatic unistrand clusters are likely a general mechanism of ERV control across *Drosophila* species and we provide evidence that multiple instances of independent evolution.

2. What is the mechanism of conversion between unistrand and dual-strand piRNA clusters? Authors should consider to conduct genetic manipulation to force the conversion in one species.

We agree that understanding how clusters emerge and/or change over evolutionary time would be interesting, however, due to the timeframe involved and major differences between germline and somatic cluster transcription, this cannot feasibly be addressed experimentally. Dual-strand clusters in *Drosophila* are non-canonically transcribed by a germline-specific machinery through Rhino-dependent recruitment of RNA Pol II to heterochromatic locations across the entire cluster region (Andersen et al., 2017; Mohn et al., 2014). Unistrand clusters, on the other hand, rely on canonical transcription through promoters and enhancers (Goriaux et al., 2014; Mohn et al., 2014).

As discussed in our manuscript, we observed that some unistrand clusters are syntenic to dual-strand clusters. Based on these observations, we propose two hypotheses, where either a dual-strand cluster could become a unistrand clusters over evolutionary time through the acquisition of promoter elements that permit somatic expression, or alternatively a unistrand cluster may lose its expression in somatic cells through the loss of a promoter, and eventually become recognized as a dual-strand clusters by Rhino through an hitherto unknown mechanism. Since this conversion would be driven by transposon invasion happening over evolutionary time (requiring numerous generations), we do not think that such a conversion can be performed in the laboratory in a reasonable timeframe. We have updated **Fig. 8** and the discussion to clarify our model.

3. In Fig. 5g, is absence/presence of flam/flam-like clusters correlate with the quality and completeness of genomes?

We thank the reviewer for pointing this out. Yes, using transposon content alone we have indeed been unable to detect flam(/-like) clusters in low-quality genome assemblies. To emphasise this, we have now indicated species with only low-quality assemblies (N50 < 1,000,000 or NN > 3000) in the figure panel (now **Fig. 7g**). We have also expanded our description of the motivation for searching for clusters based on soma-enriched piRNAs in the results section. The underlying metrics are now available in **Supplementary Table 7**.

Taken together, unistrand clusters can readily be detected based on transposon content of several high-quality assemblies. Among tested species where it was not detected initially (n=3), we were able to detect unistrand clusters based on piRNA profiling in all cases. Hence, our results indicate that most, if not all, *Drosophila* species have somatic unistrand clusters, but that some could escape detection for technical reasons depending on the used approach (TE content vs piRNA profiling). Overall, we have not observed negative examples where a unistrand cluster is not found in species for which we have soma+total piRNAs and this is now more clearly stated in the discussion.

Minor Comments and Errors that should be addressed:

1. Fig. 4f was discussed prior to Fig. 4e.

We thank the reviewer for spotting this and have swapped these two panels to refer to them in the correct order.

2. In the legend of figure 5, a mix of uppercase and lowercase letters was used.

This has been corrected.

3. In Fig. 7, the direction of loss/gain of promoter is wrong.

We thank the reviewer for this comment but do not agree with the statement. Unistrand cluster transcription is driven by RNA Pol 2 and canonical promoter regions (Goriaux et al., 2014; Mohn et al., 2014). However, dual-strand clusters typically have no promoter(s) and are transcribed by RNA Pol II via non-canonical initiation (Andersen et al., 2017; Mohn et al., 2014). Specifically, the heterochromatin protein homolog Rhino, which binds to heterochromatic regions that are decorated in the H3K9me3 mark, internally recruits transcription initiation factors to these regions and consequently results in transcription from within dual-strand piRNA clusters (Mohn et al., 2014). We have modified the model in **Fig. 8** to make this more clear.

Reviewer #2 (Remarks to the Author):

The manuscript by van Lopik et al presents a thorough study of flam and flam-like clusters in over a hundred of *Drosophila* species. This work advances our knowledge of genomic origins of gypsy-targeting piRNAs outside *D. melanogaster*. Data presented here help in understanding the evolution of piRNA transposon defense in the somatic cells of fly ovaries. I support the publication of this work in Nature Communications after the two suggestions listed below are incorporated in the revised manuscript.

-In the discussion section, when comparing the evolutionary turnover rates of germline and flam/flamlike clusters, authors conclude that flam/flamlike clusters are more conserved. A quantitative assessment would be useful here and should serve as a more compelling evidence.

Our conclusion is based on the published literature, particularly (Gebert et al., 2021) but also (Chirn et al., 2015), not our own quantification. We believe that a suitable quantitative assessment would be interesting, yet it would require annotation and verification of all dual-strand clusters in order to draw meaningful conclusions. We think that studying the conservation/divergence of dual-strand clusters is a major undertaking in itself and beyond the scope of this work.

-Presence of a promoter driving somatic expression of a locus is hypothesized by authors as a possible route to establishing a unistrand cluster. Having identified flam and flamlike clusters in so many species, authors could attempt at determining such promoter sequences, maybe even the unifying features of these regulatory elements.

We thank the reviewer for this suggestion and agree that this would be interesting to do both experimentally and computationally. We have therefore performed ATAC-seq from nine species to identify open chromatin indicative of genomic regions containing regulatory elements. These experiments, as well as soma-enriched RNA-seq, allowed us not only to determine a more likely transcription start site (TSS) of the identified unistrand clusters but also found two previously reported key regulatory elements (Inr, Ci) to fall within ATAC-seq peaks nearby the TSS. Interestingly, these elements and peaks seem conserved from *D. melanogaster* to *D. yakuba*, but were not identified in more distantly related species. We have added this analyses as new **Fig. S12 and Fig. S13** and the updated TSS coordinates to **Supplementary Table 1**.

References

- Andersen, P. R., Tirian, L., Vunjak, M., & Brennecke, J. (2017). A heterochromatin-dependent transcription machinery drives piRNA expression. *Nature*, *549*(7670), 54–59. <https://doi.org/10.1038/NATURE23482>
- Chirn, G. W., Rahman, R., Sytnikova, Y. A., Matts, J. A., Zeng, M., Gerlach, D., Yu, M., Berger, B., Naramura, M., Kile, B. T., & Lau, N. C. (2015). Conserved piRNA Expression from a Distinct Set of piRNA Cluster Loci in Eutherian Mammals. *PLoS Genetics*, *11*(11), 1–26. <https://doi.org/10.1371/journal.pgen.1005652>
- Gebert, D., Neubert, L. K., Lloyd, C., Gui, J., Lehmann, R., & Teixeira, F. K. (2021). Large *Drosophila* germline piRNA clusters are evolutionarily labile and dispensable for transposon regulation. *Molecular Cell*, *81*(19), 3965–3978.e5. <https://doi.org/10.1016/J.MOLCEL.2021.07.011>
- Goriaux, C., Desset, S., Renaud, Y., Vaury, C., & Brassset, E. (2014). Transcriptional properties and splicing of the flamenco piRNA cluster. *EMBO Reports*, *15*(4), 411–418. <https://doi.org/10.1002/EMBR.201337898>
- Mohn, F., Sienski, G., Handler, D., & Brennecke, J. (2014). The rhino-deadlock-cutoff complex licenses noncanonical transcription of dual-strand piRNA clusters in *Drosophila*. *Cell*, *157*(6), 1364–1379. <https://doi.org/10.1016/J.CELL.2014.04.031>

Reviewers' Comments:

Reviewer #1:

Remarks to the Author:

The authors addressed the majority of my concerns from the previous review. The study provides valuable resources for both piRNA and fly communities. I support its publication.

Reviewer #2:

Remarks to the Author:

The authors have addressed all the reviewers' concerns. The revised manuscript has added more data and analyses supporting authors' conclusions and now merits the publication in Nature Comms.